# *IER5*, a DNA damage response gene, is required for Notch-mediated induction of squamous cell differentiation

**Li Pan[1], Madeleine E Lemieux[2], Tom Thomas[1], Julia M Rogers[3], Colin H Lipper[3], Winston Lee[1†], Carl Johnson[1], Lynette M Sholl[1], Andrew P South[4], Jarrod A Marto[1,5], Guillaume O Adelmant[1,5], Stephen C Blacklow[3], Jon C Aster[1]\***

[1]Department of Pathology, Brigham and Women's Hospital, and Harvard Medical School, Boston, United States; [2]Bioinfo, Plantagenet, Canada; [3]Department of Biological Chemistry and Molecular Pharmacology, Blavatnik Institute, Harvard Medical School, Boston, United States; [4]Department of Dermatology and Cutaneous Biology, Sidney Kimmel Medical College, Thomas Jefferson University, Philadelphia, United States; [5]Departmentof Oncologic Pathology and Blais Proteomics Center, Dana FarberCancer Institute, HarvardMedical School, Boston, United States

**\*For correspondence:**
jaster@rics.bwh.harvard.edu

**Present address:** [†]
Departmentof Pathology, University of Michigan Medical School, Ann Arbor, United States

**Abstract** Notch signaling regulates squamous cell proliferation and differentiation and is frequently disrupted in squamous cell carcinomas, in which Notch is tumor suppressive. Here, we show that conditional activation of Notch in squamous cells activates a context-specific gene expression program through lineage-specific regulatory elements. Among direct Notch target genes are multiple DNA damage response genes, including *IER5*, which we show is required for Notch-induced differentiation of squamous carcinoma cells and TERT-immortalized keratinocytes. *IER5* is epistatic to *PPP2R2A*, a gene that encodes the PP2A B55α subunit, which we show interacts with IER5 in cells and in purified systems. Thus, Notch and DNA-damage response pathways converge in squamous cells on common genes that promote differentiation, which may serve to eliminate damaged cells from the proliferative pool. We further propose that crosstalk involving Notch and PP2A enables tuning and integration of Notch signaling with other pathways that regulate squamous differentiation.

## Introduction

Notch receptors participate in a conserved signaling pathway in which successive ligand-mediated proteolytic cleavages by ADAM10 and γ-secretase permit intracellular Notch (ICN) to translocate to the nucleus and form a Notch transcription complex (NTC) with the DNA-binding factor RBPJ and co-activators of the Mastermind-like (MAML) family (for review, see *Bray, 2016*). Outcomes of Notch activation are dose and cell-context-dependent, in part because most Notch response elements lie within lineage-specific enhancers (*Castel et al., 2013*; *Ryan et al., 2017*; *Skalska et al., 2015*; *Wang et al., 2014*). As a result, Notch-dependent transcriptional programs vary widely across cell types.

The context-dependency of outcomes produced by Notch signaling is reflected in the varied patterns of Notch mutations that are found in different cancers (for review, see *Aster et al., 2017*). In some cancers oncogenic gain-of-function Notch mutations predominate, but in human cutaneous squamous cell carcinoma (SCC) (*South et al., 2014*; *Wang et al., 2011*) loss-of-function mutations are common, early driver events, observations presaged by work showing that loss of Notch function promotes skin cancer development in mouse models (*Nicolas et al., 2003*; *Proweller et al., 2006*).

The mechanism underlying the tumor suppressive effect of Notch appears to involve its ability to promote squamous differentiation at the expense of self-renewal, a function that is operative in other squamous epithelia (*Alcolea et al., 2014*), where Notch also has tumor suppressive activities (*Agrawal et al., 2011*; *Agrawal et al., 2012*; *Loganathan et al., 2020*). In line with this idea, conditional ablation of *Notch1* in postnatal mice results in epidermal hyperplasia and expansion of proliferating basal-like cells (*Nicolas et al., 2003*; *Rangarajan et al., 2001*). Moreover, murine and human β-papilloma viruses express E6 proteins that target MAML1 and inhibit Notch function (*Meyers et al., 2017*; *Tan et al., 2012*), thereby causing epidermal hyperplasia and delayed differentiation of infected keratinocytes. Conversely, constitutively active forms of Notch enhance keratinocyte differentiation in vitro and in vivo (*Nickoloff et al., 2002*; *Rangarajan et al., 2001*; *Uyttendaele et al., 2004*).

While these studies delineate a pro-differentiation, tumor suppressive role for Notch in squamous cells, little is known about the Notch target genes that confer this phenotype. Work to date has focused on candidate genes chosen for their known activities in keratinocytes or their roles as Notch target genes in other cell types. These include *CDKN1A*/p21 (*Rangarajan et al., 2001*), which has been linked to cell cycle arrest and differentiation (*Missero et al., 1996*); *HES1,* which represses basal fate/self-renewal (*Blanpain et al., 2006*); and *IRF6*, expression of which positively correlates with Notch activation in keratinocytes (*Restivo et al., 2011*). However, dose- and time-controlled genome-wide studies to determine the immediate, direct effects of Notch activation in squamous-lineage cells have yet to be performed.

To this end, we developed and validated 2D and 3D culture models of malignant and non-transformed human squamous epithelial cells in which tightly regulated Notch activation produces growth arrest and squamous differentiation. We find that immediate, direct Notch target genes are largely keratinocyte-specific and are associated with lineage-specific NTC-binding enhancers enriched for the motifs of transcription factors linked to regulation of keratinocyte differentiation, particularly AP1. Among these targets are multiple genes previously shown to be upregulated by DNA damage and cell stress, including *IER5*, a member of the AP1-regulated immediate early response gene family (*Williams et al., 1999*). Here, we show that *IER5* is required for Notch-induced differentiation of human SCC cells and TERT-immortalized human keratinocytes, and that this requirement is abolished by knockout of the B55α regulatory subunit of PP2A, to which IER5 directly binds. Our studies provide the first genome-wide view of the effects of Notch on gene expression in cutaneous squamous carcinoma cells, highlight previously unrecognized crosstalk between Notch and DNA response genes, and point to the existence of a Notch-IER5-PP2A signaling axis that coordinates keratinocyte differentiation.

## Establishment of a conditional Notch-on SCC model

Determination of the immediate, direct effects of Notch in a model system requires tightly timed, switch-like Notch activation. This is difficult to achieve with ligands because simple addition of soluble Notch ligands does not induce signaling (*Sun and Artavanis-Tsakonas, 1997*). Methods of triggering Notch activation include plating of cells on immobilized ligands (*Varnum-Finney et al., 2000*); treatment with EDTA, which renders Notch susceptible to activating cleavages by chelating $Ca^{2+}$ and thereby destabilizing the Notch negative regulatory region (*Rand et al., 2000*); and γ-secretase inhibitor (GSI) washout, which reliably delivers a pulse of ICN in 15–30 min to the nuclei of cells expressing mutated or truncated forms of membrane-tethered Notch (*Petrovic et al., 2019*; *Ryan et al., 2017*; *Wang et al., 2014*; *Weng et al., 2006*). Plating of adherent cells on substrate coated with immobilized ligand is confounded by the need to first produce cell suspensions with trypsin and/or EDTA, which activates Notch in cells expressing Notch receptors. EDTA treatment also suffers from several limitations: (i) Notch activation is confined to a period of several minutes immediately following EDTA addition and is therefore limited in degree and duration, possibly because chelation of $Zn^{2+}$ also rapidly inactivates ADAM metalloproteases and (ii) off-target effects of EDTA, including on surface proteins that mediate cell adhesion. GSI washout is open to criticism because γ-secretase has numerous substrates in addition to Notch receptors, raising questions about specificity. However, major phenotypes induced by treatment of flies (*Micchelli et al., 2003*), mice (*van Es et al., 2005*), and humans (*Aster and Blacklow, 2012*) with GSI are all related to Notch inhibition, strongly suggesting that Notch is the dominant GSI substrate at the organismal level. In line with these observations, in prior work we have noted that cells lacking ongoing Notch signaling

show little or no change in phenotype when treated with GSI, and we therefore selected GSI wash-out to produce timed activation of Notch in cells of squamous lineage.

To create a squamous cell model in which GSI washout activates NOTCH1 (*Figure 1A*), we first engineered a cDNA encoding a mutated truncated form of NOTCH1, ΔEGF-L1596H, that cannot respond to ligand and that has a point substitution in its negative regulatory region that produces ligand-independent, γ-secretase-dependent Notch activation (*Gordon et al., 2009*; *Malecki et al., 2006*). Notably, when expressed from retroviruses *NOTCH1* alleles bearing negative regulatory region mutations like L1596H generate Notch signals that are sufficient to produce physiologic effects in hematopoietic stem cells (induction of T cell differentiation) without causing pathophysiologic effects (induction of T cell acute lymphoblastic leukemia) (*Chiang et al., 2008*). Because Notch transcription complexes appear to largely act through 'poised' enhancers primed by lineage-specific 'pioneer' transcription factors (*Falo-Sanjuan et al., 2019*), we reasoned that squamous cell carcinoma lines with loss-of-function Notch mutations and little/no ongoing Notch signaling would be an ideal context in which to identify direct downstream targets of Notch. We therefore transduced ΔEGF-L1596H into two human SCC cell lines, IC8 and SCCT2, that have biallelic inactivating mutations in *NOTCH1* and *TP53* (*Inman et al., 2018*), lesions that were confirmed by resequencing on a clinical-grade targeted exome NGS platform (summarized in *Tables 1* and *2*).

In pilot studies, we observed that the growth of IC8 and SCCT2 cells transduced with empty virus was unaffected by the presence or absence of GSI, whereas the growth of lines transduced with ΔEGF-L1596H was reduced by GSI washout (*Figure 1—figure supplement 1A, B*). GSI washout was accompanied by rapid activation of NOTCH1 (ICN1) followed by upregulation of markers of differentiation, such as involucrin (*Figure 1—figure supplement 1C, D*). Of interest, ICN1 levels reproducibly peaked at around 4 hr and then declined, suggesting that sustained NOTCH1 activation and accompanying changes in cell state led to induction of feedback loops that negatively regulate ICN1. We also observed that IC8-ΔEGF-L1596H cells formed 'skin-like' epithelia when seeded onto organotypic 3D cultures, whereas SCCT2-ΔEGF-L1596H cells did not (data not shown); therefore, additional studies focused on IC8 cells and derivatives thereof.

To further characterize and validate our system, we performed single-cell cloning of IC8−ΔEGF-L1596H cells and observed that differentiation following GSI washout correlated with ICN1 accumulation (*Figure 1—figure supplement 2A and B*). The subclone SC2, which showed moderate accumulation of ICN1 and sharply reduced growth following GSI washout, was selected for further study. The ability of SC2 cells to form a multilayered epithelium in the Notch-on, growth suppressive state appears to stem from an unexpected property that emerged in 3D cultures, namely the self-organization of these cells into a proliferating, ICN1-low basal layer in contact with matrix and a non-proliferating, ICN1-high suprabasal layer (*Figure 1—figure supplement 2C*). The self-organization of Notch-on SC2 cells grown on collagen rafts into ICN1-low basal proliferating cells and ICN1-high suprabasal non-proliferating cells suggested that contact of SC2 cells with collagen reduces ICN1 levels. To test this idea, we plated SC2 cells on plastic or collagen, washed out GSI, and compared ICN1 levels by western blotting. As shown in *Figure 1—figure supplement 2D*, culture of SC2 cells on collagen sharply reduced ICN1 levels, whereas the level of total NOTCH1 polypeptides was unchanged or slightly increased in cells grown on collagen. These observations point to the existence of one or more matrix-dependent effects that decrease ICN1 levels and which may serve to reinforce the 'Notch-low' status of basal keratinocytes.

We also noted that the growth arrest induced by GSI washout in SC2 cells was blocked by dominant-negative MAML1 (DN-MAML), a specific inhibitor of Notch-dependent transcription (*Figure 1B*; *Nam et al., 2007*; *Weng et al., 2003*), confirming that the growth inhibitory effects of GSI washout are mediated through Notch activation. Growth arrest occurred several days after Notch activation (compare *Figure 1B and C*) and was accompanied by upregulation of multiple markers of squamous differentiation, such as involucrin, keratin1, and plakophilin1, in 2D (*Figure 1C–E*) and in 3D cultures (*Figure 1F*). In addition, we also noted that staining for keratin14, a prototypic marker of proliferating basal keratinocytes (*Fuchs, 1995*), became more sharply localized to basal cells in Notch-on SC2 cells (*Figure 1F*).

Although Notch activation in IC8 and SC2 cells clearly induced expression of squamous differentiation markers, the distribution of these markers in 3D cultures failed to precisely mimic that of normal epidermis, as staining for involucrin and plakophilin-1 was seen in proliferating keratin14-positive basal cells. The observed expression of spinous markers in ICN1-low proliferating basal cells

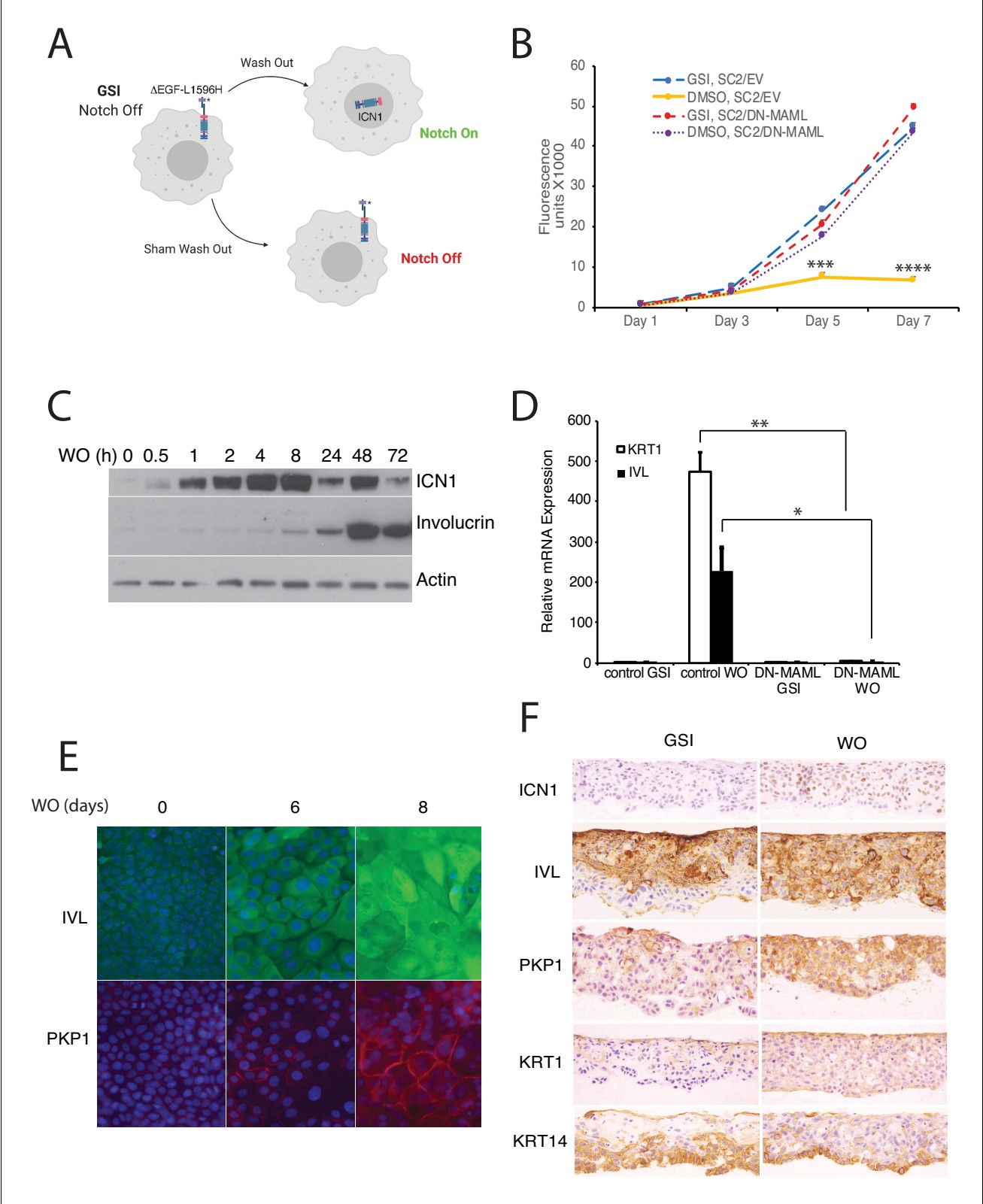

**Figure 1.** Notch activation induces growth arrest and differentiation of squamous carcinoma cells. (**A**) Strategy used to activate Notch in a tightly regulated fashion. (**B**) Notch-induced suppression of SC2 cell growth in standard cultures is abrogated by DN-MAML, a specific inhibitor of canonical Notch signaling. SC2 cells were transduced with empty MigRI virus (EV) or with MigRI virus encoding DN-MAML. Cell numbers at various times post-GSI washout (DMSO vehicle alone) or sham GSI-washout (GSI) were assessed using Cell Titer-Blue on biological replicates performed in quadruplicate.
*Figure 1 continued on next page*

*Figure 1 continued*

Error bars represent standard deviations. Timepoints with significantly different cell growth between Notch-on cells (DMSO, empty vector) and Notch-off cells (GSI, empty vector; DMSO, DN-MAML1; and GSI, DN-MAML1) are denoted with *** (p<0.005) or **** (p<0.0005) (two-tailed student t test). (C) Western blot showing the kinetics of activated intracellular NOTCH1 (ICN1) generation and increases in involucrin (IVL) following GSI washout in SC2 cells in standard cultures. (D) Notch-induced differentiation of SC2 cells is abrogated by DN-MAML. Transcripts for involucrin (IVL) and keratin1 (KRT1) were measured in the presence of GSI and 3 days after GSI washout in SC2 cells transduced with empty virus or with DN-MAML. Transcript abundance in biological replicates performed in triplicate was measured by RT-PCR and normalized against GAPDH. Error bars represent standard deviations of the mean. **, p<0.005; *, p<0.05; two-tailed student t-test. (E) Indirect immunofluorescence microscopy showing staining for involucrin (IVL, green) and plakophilin-1 (PKP1, red) in SC2 cells at time 0 and 6 and 8 days after GSI washout. Nuclei in each image were counterstained with DAPI. (F) Immunohistochemical staining of SC2 cells grown in skin raft cultures for 14 days in the presence and absence of GSI.

The online version of this article includes the following figure supplement(s) for figure 1:

**Figure supplement 1.** Notch activation induces differentiation and growth arrest of the squamous carcinoma cell lines IC8 and SCCT2.
**Figure supplement 2.** Characterization of clones derived from single IC8 cells transduced with ΔEGF-L1596H.
**Figure supplement 3.** Immunohistochemical assessment of p63, BCL6, keratin5 (KRT5), and filaggrin (FLGR) protein levels in S2 cells grown in the Notch-off (GSI) or Notch-on (WO) states in 3D rafts.

may be explained by differing Notch dose requirements for growth arrest (relatively high) versus induction of spinous differentiation (relatively low). To further compare and contrast the effects of Notch activation in SC2 cells with normal keratinocyte differentiation, we performed staining for a series of additional markers (*Figure 1—figure supplement 3*). Similar to normal keratinocyte differentiation, Notch activation was associated with increased staining for BCL6, particularly in suprabasal SC2 cells. By contrast, p63 staining, which is normally confined to basal keratinocytes, was seen in suprabasal as well as basal SC2 cells, and keratin5 staining was observed throughout SC2 cell rafts, in contradistinction to keratin14 staining (*Figure 1F*). Notch activation also failed to induce staining for markers of terminal differentiation such as filaggrin (*Figure 1—figure supplement 3*) and loricrin (not shown). Thus, Notch activation in SC2 cells induces genes associated with spinous differentiation but is insufficient to down-regulate certain genes associated with basal cell fate, such as p63 and keratin5, or to push SC2 cells to terminally differentiate.

## Identification of a squamous cell-specific Notch-induced program of gene expression

To determine early and delayed effects of Notch on gene expression, we performed RNA-seq on SC2 cells in 2D cultures in the Notch-off state and following Notch activation. Because γ-secretase has numerous substrates, as a control we performed RNA-seq on parental, non-transduced IC8 cells in the presence of GSI and following GSI washout, which revealed no significant GSI-dependent changes in gene expression in the absence of a Notch transgene (*Figure 2—figure supplement 1*). By contrast, Notch activation in SC2 cells produced significant changes in gene expression by 4 hr that became more pronounced by 24 hr and 72 hr (*Figure 2A*, see *Supplementary files 1–3* for differentially expressed genes). Gene ontology (GO) analysis revealed enrichment among upregulated genes for those that are associated with keratinocyte differentiation and biology (*Supplementary files 4* and *5*, summarized in *Figure 2B*). Among the rapidly upregulated genes (genes that increase in expression by 4 hr) were several reported targets of Notch in keratinocytes (e.g, *RHOV* [*Pomrey and Radtke, 2010*], *HES1* [*Blanpain et al., 2006*], and *IRF6* [*Restivo et al., 2011*]), but most genes in this class were novel for keratinocytes and included: (1) direct targets of Notch in other lineages (e.g. *NRARP*, *HES4*, and *HES5*); (2) genes linked to keratinocyte differentiation (e.g. *RIPK4* [*Kwa et al., 2014*], recently reported to be upregulated following treatment of keratinocytes with EDTA [*Loganathan et al., 2020*], and *SMAD3* [*Meyers et al., 2017*]); (3) genes associated with DNA damage responses in keratinocytes and other cell types (e.g. *GADD45A*, *CXCL8*, *IL1B*, *ID3*, *CYR61*, *BTG2*, *IER3*, and *IER5* [*Kis et al., 2006*; *Kumar et al., 1998*; *Maeda et al., 2002*; *Rouault et al., 1996*; *Sesto et al., 2002*; *Simbulan-Rosenthal et al., 2006*]); and (4) genes associated with growth arrest of keratinocytes and other cell types (e.g. *HES1* [*Blanpain et al., 2006*], *GADD45A* [*Maeda et al., 2002*], and *BTG2* [*Rouault et al., 1996*]).

To confirm that these changes in gene expression are general features of Notch activation in IC8-ΔEGF-L1596H cells and to determine the kinetics of response, we performed RT-PCR analyses on a number of known and novel targets in pooled IC8-ΔEGF-L1596H transductants. This confirmed the

**Table 1.** Sequence variants, IC8* and SCCT2** squamous cell carcinoma cell lines.

| Gene | Variant | Variant allele frequency |
|---|---|---|
| **IC8 cell** | | |
| CASP8 | c.971T > C(p.M324T) | 66% of 411 reads |
| FBXW7 | c.1633T > C(p.Y545H) | 33% of 195 reads |
| KMT2D | c.7412G > A(p.R2471Q) | 42% of 255 reads |
| MGA | c.5599G > A(p.V1867I) | 39% of 710 reads |
| MTOR | c.4828G > A(p.E1610K) | 56% of 280 reads |
| NOTCH1 | c.5059C > T (p.Q1687*) | 100% of 412 reads |
| PAXIP1 | c.2023C > T(p.H675Y) | 17% of 384 reads |
| PMS1 | c.566_567delTCinsAT(p.V189D) | 36% of 108 reads |
| RIF1 | c.658G > A(p.E220K) | 62% of 251 reads |
| ROS1 | c.1144T > G(p.Y382D) | 87% of 169 reads |
| ROS1 | c.1164+2_1164+8delTTAGTCC () | 19% of 191 reads |
| SDHA | c.1627T > C(p.Y543H) | 56% of 668 reads |
| SF3B1 | c.2549T > C(p.I850T) | 31% of 246 reads |
| TERT | CC242-243TT promoter mutation | 50% of 26 reads |
| TP53 | c.451C > T(p.P151S) | 100% of 366 reads |
| WHSC1 | c.2185C > T(p.R729C) | 66% of 410 reads |
| WWTR1 | c.551T > G (p.V184G) | 64% of 256 reads |
| ZNF217 | c.2590C > T(p.L864F) | 39% of 835 reads |
| ZNF217 | c.1162delC(p.H388Tfs*77) | 55% of 822 reads |
| **SCCT2 Cell** | | |
| ALK | c.2854G > A (p.G952R) | 50% of 441 reads |
| ASXL1 | c.3959C > T (p.A1320V) | 31% of 930 reads |
| BRD3 | c.533C > T (p.S178F) | 49% of 281 reads |
| BRD4 | c.3915_3917dupTGC (p.A1306dup) | 45% of 170 reads |
| CDH4 | c.1801C > T (p.L601F) | 30% of 447 reads |
| CDKN2A | c.*151–1G > A () | 100% of 172 reads |
| CDKN2A | c.212A > T (p.N71I) | 100% of 184 reads |
| CREBBP | c.5842C > T (p.P1948S) | 74% of 77 reads |
| CREBBP | c.2116G > A (p.G706R) | 45% of 172 reads |
| DDB1 | c.327+6G > A () | 47% of 451 reads |
| DICER1 | c.775C > T (p.P259S) | 42% of 301 reads |
| DOCK8 | c.185T > A (p.V62E) | 100% of 597 reads |
| EGFR | c.1955G > A (p.G652E) | 48% of 518 reads |
| EGFR | c.298C > T (p.P100S) | 49% of 595 reads |
| ERCC2 | c.886A > T (p.S296C) | 48% of 165 reads |
| ERCC5 | c.264+1G > A () | 50% of 442 reads |
| ETV4 | c.1298C > G (p.P433R) | 45% of 302 reads |
| FANCF | c.494C > T (p.T165I) | 50% of 644 reads |
| FANCL | c.155+1G > A () | 51% of 220 reads |
| FAT1 | c.9076–1G > A () | 49% of 367 reads |
| FH | c.681G > T (p.Q227H) | 4% of 756 reads |
| FLT4 | c.2224G > A (p.D742N) | 49% of 346 reads |
| GALNT12 | c.1035+5G > A () | 52% of 523 reads |
| GLI2 | c.1859C > A (p.T620K) | 45% of 351 reads |

*Table 1 continued on next page*

*Table 1 continued*

| Gene | Variant | Variant allele frequency |
|------|---------|--------------------------|
| HNF1A | c.1640C > T (p.T547I) | 54% of 392 reads |
| JAZF1 | c.477C > T (p.I159I) | 47% of 606 reads |
| JAZF1 | c.328C > T (p.P110S) | 44% of 211 reads |
| KMT2D | c.10355+1G > A () | 49% of 622 reads |
| LIG4 | *c.1271_1275delAAAGA (p.K424Rfs*20)* | 40% of 659 reads |
| MAP2K1 | c.568+1G > A () | 53% of 239 reads |
| MED12 | c.2080G > A (p.E694K) | 100% of 269 reads |
| MYB | c.1461+5G > A () | 41% of 430 reads |
| NF1 | c.2608G > A (p.V870I) | 50% of 615 reads |
| NF2 | c.813T > G (p.F271L) | 48% of 168 reads |
| NOTCH1 | c.1226G > T (p.C409F) | 44% of 519 reads |
| NOTCH1 | c.1406A > G (p.D469G) | 50% of 912 reads |
| NOTCH1 | c.1245G > T (p.E415D) | 42% of 495 reads |
| NOTCH2 | c.5252G > A (p.G1751D) | 44% of 459 reads |
| NOTCH2 | c.1298G > A (p.C433Y) | 50% of 484 reads |
| NOTCH2 | c.1108+1G > A () | 53% of 305 reads |
| NSD1 | c.7669G > A (p.G2557R) | 49% of 743 reads |
| PDGFRB | c.2586+2T > A () | 43% of 380 reads |
| PHOX2B | c.181A > T (p.T61S) | 52% of 222 reads |
| POLQ | c.6565G > A (p.A2189T) | 27% of 462 reads |
| POLQ | c.1634G > A (p.S545N) | 33% of 667 reads |
| PPARG | c.819+6T > C () | 100% of 134 reads |
| PRKDC | c.6436G > A (p.A2146T) | 42% of 471 reads |
| RAD51C | c.996G > A (p.Q332Q) | 45% of 302 reads |
| RHEB | c.443C > T (p.S148F) | 46% of 120 reads |
| ROS1 | c.6871C > T (p.P2291S) | 45% of 605 reads |
| ROS1 | c.3342A > T (p.Q1114H) | 48% of 274 reads |
| ROS1 | c.137A > T (p.D46V) | 42% of 215 reads |
| RPTOR | c.2992G > A (p.V998I) | 48% of 352 reads |
| RUNX1T1 | c.1039G > A (p.D347N) | 45% of 715 reads |
| SDHA | c.1151C > T (p.S384L) | 53% of 446 reads |
| SLC34A2 | c.1700T > A (p.I567N) | 50% of 460 reads |
| SMARCA4 | c.3947T > G (p.F1316C) | 55% of 431 reads |
| SMARCE1 | c.395C > T (p.A132V) | 51% of 587 reads |
| STAT3 | c.1852G > A (p.G618S) | 47% of 527 reads |
| TDG | c.166+4G > A () | 48% of 329 reads |
| TP53 | c.375+1G > T | 47% of 173 reads |
| TP53 | c.832_833delCCinsTT (p.P278F) | 46% of 418 reads |
| UIMC1 | c.971T > C (p.V324A) | 48% of 745 reads |
| XPC | c.571C > T (p.R191W) | 100% of 219 reads |

[*]Based on analysis of 16,131,317 unique, high-quality sequencing reads (mean, 406 reads per targeted exon, with 98% of exons having more than 30 reads).

[†]Based on analysis of 20,972,158 unique, high-quality sequencing reads (mean, 413 reads per targeted exon, with 99% of exons having more than 30 reads).

Notch responsiveness of all genes tested, and also revealed variation in the kinetics of response, even among 'canonical' Notch target genes. For example, *HES1* showed fast induction followed by rapid down-regulation, consistent with autoinhibition (*Hirata et al., 2002*), whereas *HES4*, *HES5*, and *NRARP* (a feedback inhibitor of NTC function [*Jarrett et al., 2019*]) showed more sustained increases in expression (*Figure 2C*). Genes encoding non-structural proteins known to be linked to squamous differentiation also were 'early' responders (*Figure 2D*), as were genes linked to DNA damage/cell stress response (*Figure 2E*). In the case of the latter novel targets, we confirmed that protein levels also rose in a Notch-dependent fashion (*Figure 2F*). By contrast, increased expression of genes encoding structural proteins associated with keratinocyte differentiation (e.g. *IVL*, *KRT1*, *KRT13*) was delayed, only emerging at 24–72 hr (*Supplementary files 1–3*). These findings suggest that Notch activation induces the expression of a core group of early direct target genes, setting in motion downstream events that lead to differentiation.

Notch activation also down-regulated a smaller set of genes (*Figure 2A*, summarized in *Supplementary files 1–3*), possibly via induction of transcriptional repressors of the Hes family. These include multiple genes expressed by basal epidermal stem cells, including genes encoding the Notch ligand DLL1 (*Lowell et al., 2000*); β1-integrin (*Jones and Watt, 1993*); LRIG1 (*Jensen and Watt, 2006*), a negative regulator of epidermal growth factor receptor signaling; and multiple WNT ligands (WNT7A, 7B, 9A, 10A, and 11), of interest because WNT signaling contributes to maintenance of epidermal stem cells (*Lim et al., 2013*).

To determine the overlap of Notch target genes in squamous cells with other cell lineages, we compared the list of Notch-responsive genes in SC2 cells with three other cell types in which GSI washout has been used to identify genes that are rapidly upregulated by Notch: triple-negative breast cancer cells (*Petrovic et al., 2019*); mantle cell lymphoma cells (*Ryan et al., 2017*); and T-cell acute lymphoblastic leukemia (T-ALL) cells. Using fairly stringent cutoffs for Notch-responsiveness (FDR < 0.05, log2 change >1; summarized in *Supplementary file 6*), we failed to identify any genes that were co-regulated by Notch in all of these cell types (*Figure 2—figure supplement 2*). Even in two epithelial cell types, SC2 squamous cells and MB157 triple negative breast cancer cells, only 10.8% of Notch-responsive genes in SC2 cells were also Notch-responsive in MB157 cells. As would be expected, the overlap between Notch-responsive genes in SC2 cells and B lineage REC1 cells (9/390, 2.3%) and T lineage DND41 cells (1/390, 0.3%) was even lower. These observations serve to again emphasize the remarkable context-specificity of Notch effects on gene expression.

## Notch target genes are associated with lineage-specific NTC-binding enhancer elements

To identify sites of NTC-binding to Notch-responsive regulatory elements in IC8-ΔEGF-L1596H cells, we performed ChIP-seq for Notch transcription complex components (RBPJ and MAML1) 4 hr after Notch activation, as well as for RBPJ prior to Notch activation. The rationale for identifying RBPJ and MAML1-binding sites, rather than NOTCH1-binding sites, was several fold: (i) we wanted to identify binding of endogenous Notch transcription complex components with reliable commercially available monoclonal antibodies; (ii) based on our RNA-seq data sets, *MAML1* is the most highly expressed member of the MAML family in IC8 cells (MAML1 log$_2$ read counts per million = 6.66; MAML2 log$_2$ read counts per million = 3.59; MAML3 log$_2$ read counts per million = 0.70), and therefore was the logical member of the family to study; and (iii) we were intrigued by prior studies suggesting that MAML1 might associate with non-Notch transcription factor complexes (*Jin et al., 2010*; *Quaranta et al., 2017*; *Shen et al., 2006*; *Zhao et al., 2007*). In the Notch-on state, we found that most MAML1 binding sites also bound RBPJ (8533/9,187 sites, 93%; *Figure 3A*), in line with studies showing that MAML1 association with DNA requires both RBPJ and NICD (*Nam et al., 2006*). Approximately 92% of RBPJ/MAML1 co-binding sites (hereafter designated NTC binding sites) are in intergenic or intronic regions consistent with enhancers (*Figure 3B*). As predicted by past studies (*Castel et al., 2013*; *Krejcí and Bray, 2007*; *Ryan et al., 2017*; *Wang et al., 2014*), NTC binding was associated with increases in RBPJ ChIP-Seq signals and H3K27ac signals at promoter and enhancer sites (*Figure 3C*), features previously noted to characterize 'dynamic' functional Notch response elements. Motif analysis revealed that the most common motif lying within 300 bp of NTC ChIP-Seq signals is that of RBPJ (*Figure 3D*), and that the motif for AP1, a factor not associated with NTC binding sites in other cell types (*Chatr-Aryamontri et al., 2017*; *Drier et al., 2016*; *Petrovic et al., 2019*; *Ryan et al., 2017*; *Wang et al., 2014*), is also highly enriched in this 600 bp

**Table 2.** Copy number variants, squamous cell carcinoma cell lines.

| Chromosome | Type | Genes affected |
|---|---|---|
| **IC8 cell line** | | |
| 1q | Gain | MCL1, GBA, RIT1, NTRK1, DDR2, PVRL4, SDHC, CDC73, MDM4, PIK3C2B, UBE2T, PTPN14, H3F3A, EGLN1, AKT3, EXO1, FH |
| 2 | Loss | XPO1, FANCL, REL, MSH6, EPCAM, MSH2, SOS1, ALK, BRE, DNMT3A, GEN1, MYCN, TMEM127, GLI2, ERCC3, CXCR4, RIF1, ACVR1, ABCB11, NFE2L2, PMS1, CASP8, SF3B1, CTLA4, ERBB4, IDH1, BARD1, XRCC5, DIS3L2 |
| 3p | Loss | MITF, BAP1, PBRM1, COL7A1, RHOA, SETD2, CTNNB1, MLH1, MYD88, XPC, PPARG, RAF1, FANCD2, OGG1, VHL |
| 3q | Gain | NFKBIZ, CBLB, POLQ, GATA2, MBD4, TOPBP1, FOXL2, ATR, MECOM, PRKCI, TERC, PIK3CA, SOX2, ETV5, BCL6 |
| 4 | Loss | PHOX2B, RHOH, SLC34A2, FGFR3, WHSC1, KDR, KIT, PDGFRA, FAM175A, HELQ, TET2, FBXW7, NEIL3, FAT1 |
| 5 | Gain | RICTOR, IL7R, SDHA, TERT, MAP3K1, PIK3R1, XRCC4, RASA1, APC, RAD50, CTNNA1, PDGFRB, ITK, NPM1, TLX3, FGFR4, NSD1, UIMC1, FLT4 |
| 6 | Gain | CCND3, NFKBIE, POLH, VEGFA, CDKN1A, PIM1, RNF8, FANCE, DAXX, HFE, HIST1H3B, HIST1H3C, ID4, PRDM1, ROS1, RSPO3, MYB, TNFAIP3, ESR1, ARID1B, PARK2, QKI |
| 7 | Gain | EGFR, IKZF1, JAZF1, ETV1, PMS2, RAC1, CARD11, SBDS, CDK6, SLC25A13, CUX1, RINT1, MET, POT1, SMO, BRAF, PRSS1, EZH2, RHEB, XRCC2, PAXIP1 |
| 8p | Loss | KAT6A, POLB, FGFR1, WHSC1L1, NRG1, WRN, NKX3-1, PTK2B, GATA4, NEIL2 |
| 8q11.21-q21.11 | Loss | PRKDC, MYBL1, TCEB1 |
| 8q21.3-q24.3 | Gain | NBN, RUNX1T1, RAD54B, RSPO2, EXT1, RAD21, MYC, RECQL4 |
| 9p13.2-p21.3 | Loss | PAX5, FANCG, RMRP, CDKN2A, CDKN2B, MTAP |
| 9p24.1-p24.3 | Gain | CD274, JAK2, PDCD1LG2, DOCK8 |
| 11p11.2-p13 | Gain | EXT2, LMO2 |
| 13q33.1 | Loss | ERCC5 |
| 15q | Gain | FAN1, GREM1, BUB1B, MGA, RAD51, TP53BP1, B2M, USP8, MAP2K1, PML, NEIL1, FAH, NTRK3, BLM, FANCI, IDH2, IGF1R |
| 16p13.3 | Loss | CREBBP, SLX4 |
| 19 | Loss | BABAM1, CRTC1, JAK3, KLF2, MEF2B, BRD4, NOTCH3, CALR, KEAP1, SMARCA4, ELANE, GNA11, MAP2K2, STK11, TCF3, CCNE1, C19orf40, CEBPA, AKT2, AXL, CIC, XRCC1, ARHGAP35, ERCC1, ERCC2, BCL2L12, PNKP, POLD1, PPP2R1A |
| 20 | Gain | MCM8, ASXL1, BCL2L1, MAFB, AURKA, ZNF217, GNAS, CDH4 |
| **SCCT2 Cell Line** | | |
| 1q32.1 | Loss | UBE2T |
| 1q42.12-q42.2 | Gain | H3F3A, EGLN1 |
| 1q43 | Loss | AKT3, EXO1 |
| 1q43 | Gain | FH |

*Table 2 continued on next page*

Table 2 continued

| Chromosome | Type | Genes affected |
|---|---|---|
| 3 p Arm level | Loss | *MITF, BAP1, PBRM1, COL7A1, RHOA, SETD2, CTNNB1, MLH1, MYD88, XPC, PPARG, RAF1, FANCD2, OGG1, VHL* |
| 3q Arm level | Gain | *NFKBIZ, CBLB, POLQ, GATA2, MBD4, TOPBP1, FOXL2, ATR, MECOM, PRKCI, TERC, PIK3CA, SOX2, ETV5, BCL6* |
| 8q Arm level | Gain | *PRKDC, MYBL1, TCEB1, NBN, RUNX1T1, RAD54B, RSPO2, EXT1, RAD21, MYC, RECQL4* |
| 9q Arm level | Gain | *GNAQ, NTRK2, FANCC, PTCH1, GALNT12, XPA, KLF4, TAL2, ENG, ABL1, TSC1, BRD3, NOTCH1* |
| 18q11.2 | Gain | *GATA6, RBBP8* |
| 18q11.2-q21.33 | Gain | *SS18, SETBP1, SMAD2, SMAD4, BCL2* |
| 20 | Gain | *MCM8, ASXL1, BCL2L1, MAFB, AURKA, ZNF217, GNAS, CDH4* |

window. Based on the method of *Severson et al., 2017*, approximately 13% of NTC-binding sites in IC8 cells are predicted to be sequence paired sites (*Figure 3E*), a specialized type of response element that binds NTC dimers (*Arnett et al., 2010*). Finally, particularly at early time points, NTC-binding sites were spatially associated with genes that are upregulated by Notch, whereas genes that decreased in expression were no more likely to be associated with NTC binding sites than genes that did not change in expression (*Figure 3F*). Taken together, these studies show that NTCs mainly bind lineage-specific enhancers in SCC cells and that their loading leads to rapid 'activation' of Notch-responsive elements and upregulation of adjacent genes.

We also performed motif analysis on sites producing significant signals for only RBPJ or only MAML1. RBPJ 'only' sites also were enriched for RBPJ (E value 1.3$^{e-124}$) and AP1 (E value 2.8$^{e-71}$) motifs but had lower average ChIP-Seq signals, suggesting these may be weak RBPJ-binding sites. MAML1 'only' sites also were enriched for AP1 motifs (E value 2.9$^{e-181}$) but were not associated with RBPJ motifs. These sites were relatively few in number (N = 654) and the associated AP1 motifs were distributed broadly around MAML1 signal peaks, arguing against direct physical interaction between MAML1 and AP1 family members on chromatin. Thus, the significance of these 'MAML1-only' peaks is uncertain, and it is possible that the observed ChIP-seq signals are non-specific, stemming from over-representation of 'open' chromatin in ChIPs.

## *IER5* is a direct Notch target gene

We were intrigued by the convergence of Notch target genes, which presumably serve to promote and coordinate keratinocyte differentiation, and genes linked to DNA damage/cell stress responses. We selected one gene of this class, *IER5*, a member of the immediate early response gene family, for detailed analysis based on prior work implicating *IER5* in cellular responses to DNA damaging agents and heat shock (*Ding et al., 2009*; *Ishikawa and Sakurai, 2015*; *Kis et al., 2006*), as well as functional studies suggesting that IER5 is a modulator of the serine/threonine kinase PP2A (*Asano et al., 2016*; *Ishikawa et al., 2015*; *Kawabata et al., 2015*) and might therefore serve as point of crosstalk between Notch and signaling pathways that depend on serine/threonine phosphorylation for signal transduction.

To confirm that *IER5* is also upregulated by activation of endogenous Notch signaling in non-transformed keratinocytes, we studied TERT-immortalized NOK1 keratinocytes, which undergo squamous differentiation when moved to high $Ca^{2+}$ medium (*Piboonniyom et al., 2003*). We observed that differentiation of NOK1 cells significantly increased the expression of *IER5* as well as the canonical Notch target gene *NRARP*, effects that were blocked by GSI and by DN-MAML1, confirming that the observed changes in gene expression are Notch-dependent (*Figure 4—figure supplement 1A and B*). In line with the Notch-dependent increases in *IER5* transcripts upon induction of differentiation, we also observed that IER5 protein levels increased in differentiation medium in a GSI- and DN-MAML1-sensitive fashion (*Figure 4—figure supplement 1C and D*). Differentiation was

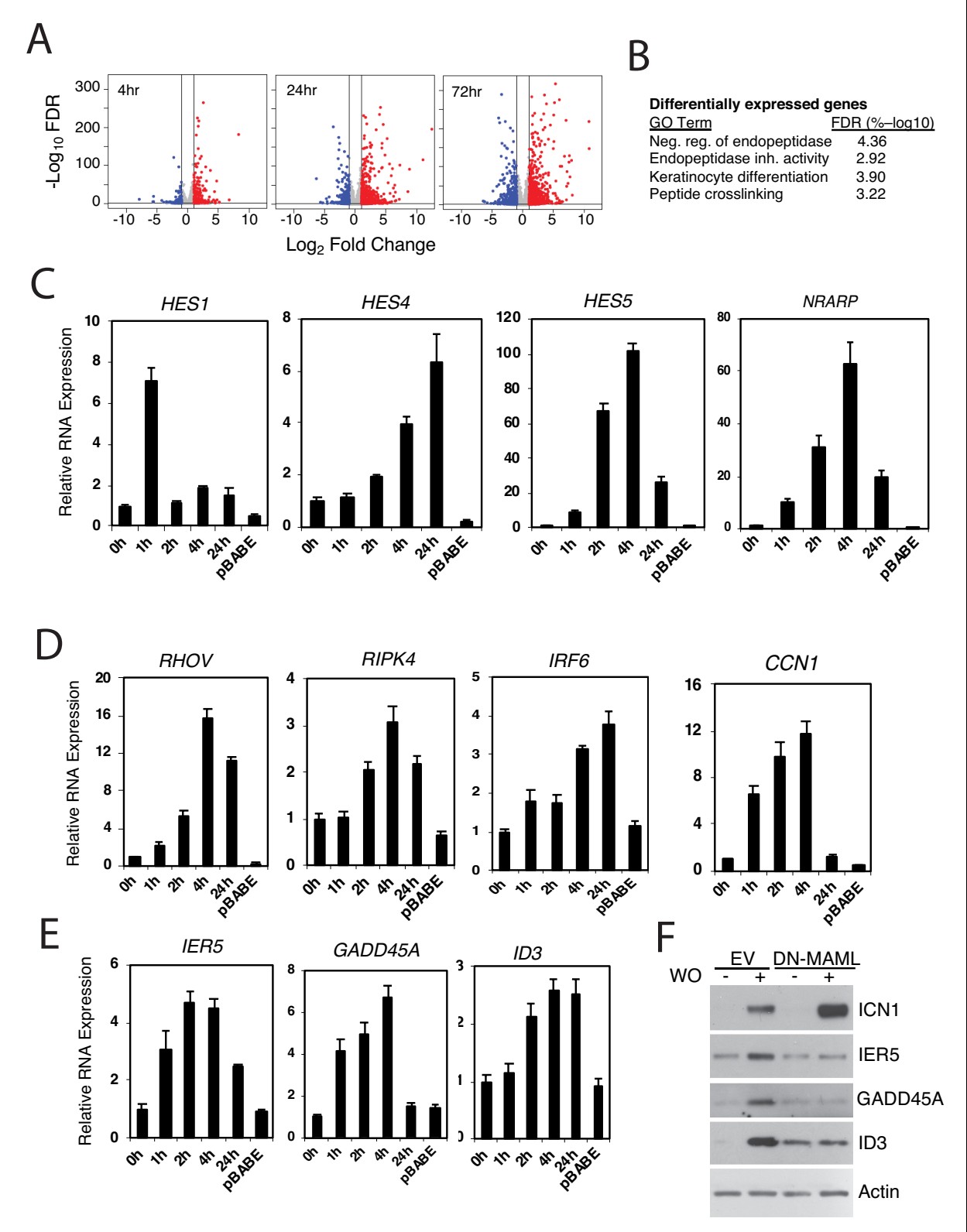

**Figure 2.** Identification of Notch-induced genes in squamous carcinoma cells. (**A**) Volcano plots showing changes in RNA transcript read counts induced by Notch activation in SC2 cells for 4, 24, and 72 hr as compared to control cells treated with sham GSI washout. RNA-seq for each treatment group was performed in triplicate on biological replicates. Vertical lines denote a twofold change in read count, while the horizontal line denotes a false discovery rate (FDR) of 5%. (**B**) Gene ontogeny (GO) annotation of differentially expressed genes in 'Notch-on' SC2 cells. The most highly associated
*Figure 2 continued on next page*

*Figure 2 continued*

GO terms are shown; other significant associated annotated gene sets (FDR < 5%) are listed in *Supplementary file 7*. (C-E) Transcriptional responses of selected 'canonical' Notch target genes (C), genes linked to keratinocyte differentiation (D), and genes associated with DNA damage responses (E), to Notch activation in IC8-ΔEGF-L1596H cells. Transcript abundance in technical replicates prepared in triplicate was measured by RT-PCR and normalized against GAPDH. Error bars represent standard deviations of the mean. (F) Western blots of cell lysates prepared from IC8-ΔEGF-L1596H cells transduced with empty virus (EV) or DN-MAML following sham GSI washout (-) or 24 hr post-GSI washout (+).

The online version of this article includes the following figure supplement(s) for figure 2:

**Figure supplement 1.** GSI treatment has little effect on gene expression in IC8 squamous carcinoma cells.
**Figure supplement 2.** Venn diagram showing the overlap in Notch target genes (defined as log2 change >1 and FDR < 0.05 following GSI washout) in IC8 squamous carcinoma cells (S2 subclone), MB157 triple negative breast carcinoma cells (*Petrovic et al., 2019*), REC1 mantle cell lymphoma cells (*Ryan et al., 2017*), and DND-41 T-ALL cells (*Petrovic et al., 2019*).

accompanied by activation of NOTCH2 and NOTCH3 (inferred from the accumulation of smaller polypeptides consistent with ADAM cleavage products under differentiation conditions in the presence of GSI; *Figure 4—figure supplement 1C and D*), as well as increased expression of NOTCH3, a known target of activated Notch. Suppression of *IER5* transcript levels by GSI in some experiments was less pronounced than the abrogation of the accumulation of IER5 protein by Notch inhibitors, suggesting that additional pathways influence *IER5* expression (consistent with the complex enhancer landscape around this gene) and that Notch signaling may regulate *IER5* both transcriptionally and post-transcriptionally. Unexpectedly, we did not observe any activation of NOTCH1 in NOK1 cells in differentiation medium (data not shown), suggesting that differentiation in this model is directed by NOTCH2 instead of NOTCH1. Notably, given that germline swaps of the coding sequences of the intracellular domains of NOTCH1 and NOTCH2 yield apparently normal mice (*Liu et al., 2015*), it is likely that NOTCH1 and NOTCH2 activate the same sets of target genes, including *IER5* in keratinocytes.

We next sought to confirm that *IER5* is a direct Notch target gene. Inspection of chromatin landscapes around *IER5* in IC8 cells revealed a series of flanking enhancers, two of which (D and E) showed the largest RBPJ/MAML1 signals and the greatest increase in H3K27ac following Notch activation (*Figure 4A*), a dynamic change that is strongly correlated with increased transcription of flanking genes (*Wang et al., 2014*). Notably, a similar enhancer landscape exists in non-transformed human keratinocytes (*Figure 4—figure supplement 1E*), and expression of *IER5* transcripts is readily detectable in normal human skin (*Figure 4—figure supplement 1F-H*), consistent with the idea that the observed enhancers are involved in physiologic regulation of *IER5* in keratinocytes. Reporter gene assays with enhancers D and E in IC8-ΔEGF-L1596H cells (*Figure 4B and C*, respectively) confirmed that the Notch responsiveness of these elements depend on RBPJ-binding sites and also showed, in the case of enhancer D, that a flanking AP1 consensus site is also required. To determine the contributions of enhancers D and E within the genomic *IER5* locus, we used CRISPR/Cas9 targeting to delete the regions containing RBPJ-binding sites in these two enhancers in SC2 cells (*Figure 4D*). These deletions partially abrogated the Notch-dependent increase in *IER5* transcription (*Figure 4E*) and suppressed the accumulation of IER5 protein following Notch activation (*Figure 4F*), confirming that *IER5* is directly regulated by Notch through these elements.

## IER5 is required for 'late' Notch-dependent differentiation events in squamous cells

To systematically determine the contribution of *IER5* to Notch-dependent changes in gene expression, we compared the transcriptional response to Notch activation in SC2 cells, SC2 cells in which *IER5* was knocked out (I5 cells), and I5 cells to which *IER5* expression was added back (I5AB cells, *Figure 5A*). Different doses of *IER5* had no effect on gene expression in the absence of Notch signaling, or on the expression of genes that are induced by Notch within 4 hr (*Figure 5B*, denoted with a blue box); however, by 24 hr and 72 hr of Notch activation, I5 cells failed to upregulate a large group of Notch-responsive genes that were rescued by add-back of IER5 (*Figure 5B*, denoted with a red box). GO analysis revealed that *IER5*-dependent genes were associated with various aspects of keratinocyte differentiation and biology (*Figure 5C*; summarized in *Supplementary files 7* and *8*). An example of a differentiation-associated gene impacted by loss of *IER5* is *KRT1*, a marker of spinous differentiation, expression of which is markedly impaired by *IER5* knockout and restored by

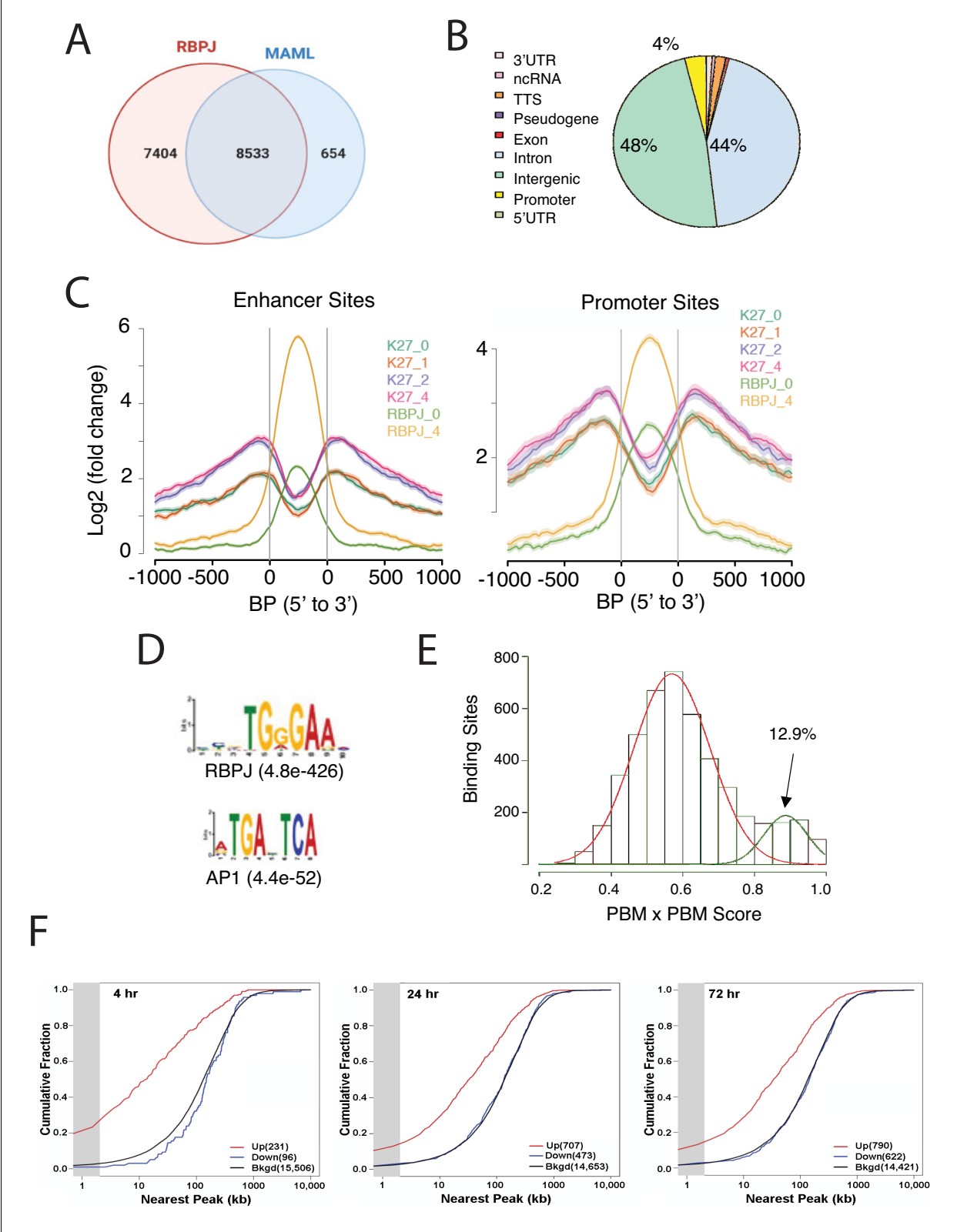

**Figure 3.** Characterization of Notch transcription complex (NTC) binding sites in IC8-ΔEGF-L1596H cells. (**A**) Number and overlap of RBPJ and MAML1 binding sites determined by ChIP-Seq of chromatin prepared 4 hr after Notch activation. (**B**) Genomic distribution of RBPJ/MAML1 co-binding sites 4 hr after Notch activation. TTS, transcription termination sites; ncRNA, non-coding RNA. (**C**) Effect of NTC loading on histone3 lysine27 acetylation (H3K27ac), based on ChIP-Seq for H3K27ac in cells maintained in GSI and in cells 1, 2, and 4 hr after GSI washout. (**D**) Transcription factor motifs

*Figure 3 continued on next page*

Figure 3 continued

enriched within 300 bp of RBPJ/MAML1 ChIP-Seq signal peaks. (E) Protein-binding matrix (PBM) X PBM scores for NTC-binding sites. Sites with scores in the right-hand Gaussian distribution correspond to likely sequence paired sites. (F) Kolmogorov-Smirnov analysis showing spatial relationships between NTC-binding sites and transcriptional start sites (TSSs) of genes that increase, decrease, or are unchanged in expression following Notch activation. The gray zone denotes genes with TSSs within 2 kb of RBP/MAML1 peaks.

*IER5* add-back (*Figure 5D*). Similarly, *IER5* was required for Notch-dependent expression of the late marker involucrin in 3D cultures (*Figure 5G*). Thus, *IER5* is necessary but not sufficient for expression of a group of genes that respond to Notch with delayed kinetics.

To extend these observations to non-transformed keratinocytes that rely on endogenous Notch signaling for differentiation, we targeted *IER5* with CRISPR/Cas9 and also enforced expression of *IER5* through retroviral transduction in NOK1 cells. Although CRISPR/Cas9 targeting of bulk NOK1 cells was only partially effective, it was sufficient to diminish the expression of multiple differentiation-associated genes (*Figure 5F*), whereas overexpression of *IER5* increased expression of each of these markers (*Figure 5G*). Thus, *IER5* is required for Notch-dependent differentiation of malignant and non-transformed keratinocytes.

## IER5 binds B55α/PP2A complexes

*IER5* encodes a 327 amino acid protein with a ~ 50 amino acid N-terminal IER domain and a C-terminal domain predicted to be unstructured, suggesting that it functions through protein-protein interactions. To identify interacting proteins in an unbiased way, we expressed a tagged form of IER5 in *IER5* null I5 cells and performed affinity purification followed by mass spectrometry (*Adelmant et al., 2019*), which identified the B55α regulatory subunit of PP2A (encoded by the *PPP2RA2* gene) and PP2A scaffolding and catalytic subunits as potential interactors (*Figure 6A*; summarized in *Supplementary file 9*). We confirmed these associations by expressing tagged IER5 in I5 cells and tagged B55α in SC2 cells (*Figure 6B*). Full-length IER5 and the N-terminal IER domain of IER5 co-precipitated endogenous B55α in Notch-independent fashion, whereas the C-terminal portion of IER5 did not (*Figure 6C*). Similarly, tagged B55α co-precipitated endogenous IER5 in a fashion that was augmented by Notch activation (*Figure 6D*), consistent with increased recovery of IER5 due to induction of *IER5* expression by Notch. To confirm that IER5 binds B55α directly, we studied the interaction of purified recombinant proteins. IER5 exhibited saturable binding to B55α-coated beads (*Figure 6E*), and additional microscale thermophoresis studies showed that IER5 binds B55α with a Kd of approximately 100 nM (*Figure 6F*).

## *IER5* is epistatic to *PPP2R2A* in SCC cells

To gain insight into the role of IER5-B55α interaction in the regulation of Notch- and *IER5*-sensitive genes, we prepared SCC cells that were knocked out for *IER5*, *PPP2R2A*, or both genes (*Figure 7A*). Knockout of *PPP2R2A* did not affect the levels of ICN1 following GSI washout (*Figure 7A*), but markedly increased the expression of the late differentiation gene *KRT1* in 2D culture (*Figure 7B*) and the accumulation of involucrin in 3D cultures (*Figure 7C*), effects that were suppressed by add-back of B55α, suggesting a model in which IER5 suppresses a B55α-dependent activity. This was supported by assays performed with *IER5*/*PPP2R2A* double knockout cells (*Figure 7A*), in which expression of the late genes such as *KRT1* was also restored (*Figure 7D*). These results suggest that Notch modulation of B55α-PP2A activity via IER5 is important in regulating the complex series of events downstream of Notch that lead to squamous cell differentiation.

## Discussion

Our work provides a genome-wide view of the direct effects of Notch in SCC cells, in which Notch activation induces growth arrest and differentiation. The phenotypic changes induced by Notch are mediated by a largely squamous-cell-specific transcriptional program that includes genes linked to keratinocyte differentiation and DNA damage responses, including *IER5*, which modulates the activity of B55α-containing PP2A complexes. Upregulation of these Notch-responsive genes are associated with binding of NTCs to RBPJ sites within lineage-specific enhancers, including a minority of sequence-paired sites, a specialized dimeric NTC binding element recently implicated in anti-

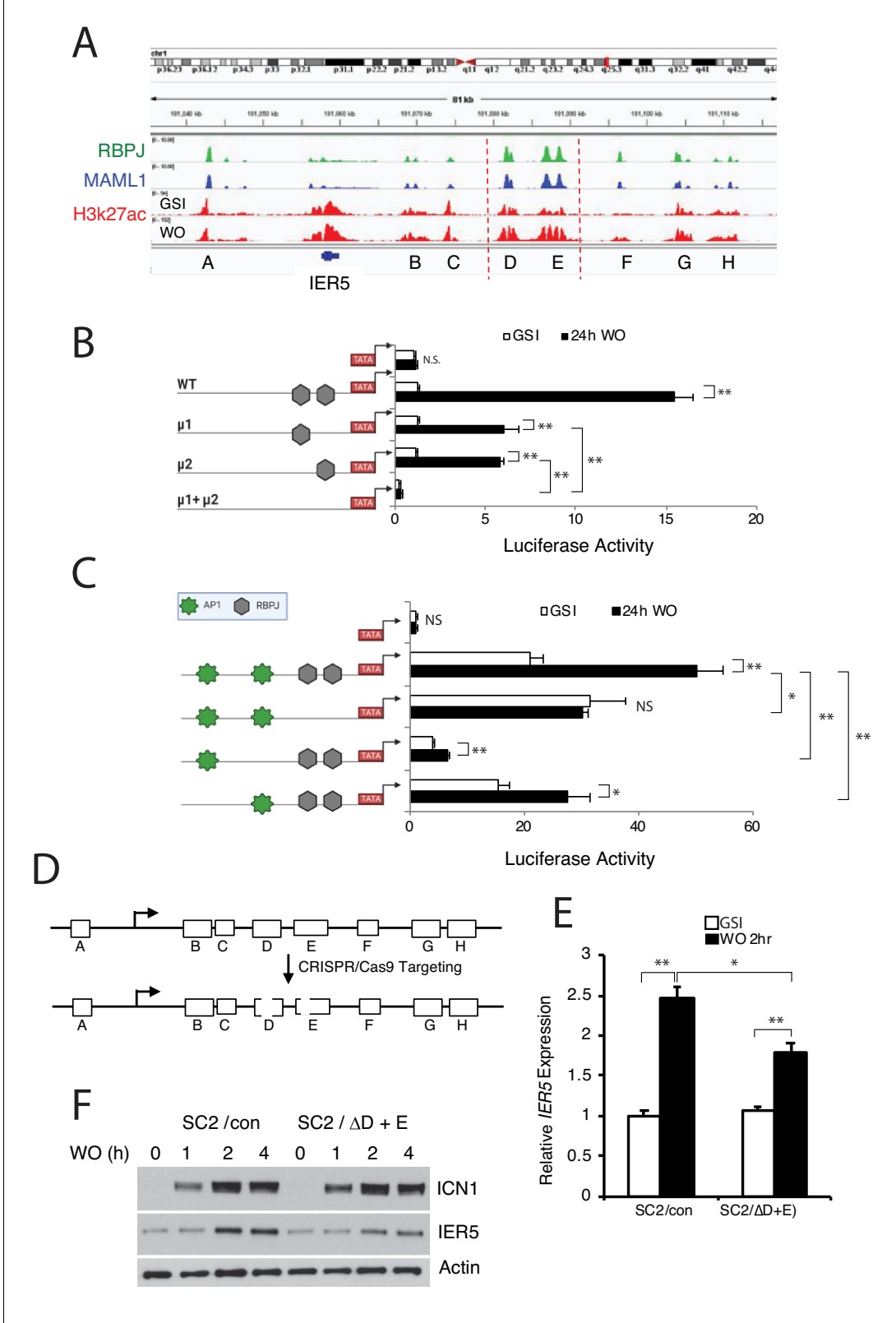

**Figure 4.** *IER5* is a direct Notch target gene. (**A**) Chromatin landscapes around *IER5* in IC8-ΔEGF-L1596H cells. ChIP-Seq signals for RBPJ, MAML1, and H3K27ac for cells maintained in and 4 hr after GSI washout (WO) are shown. (**B, C**) Activities of a WT *IER5* enhancer E luciferase reporter gene and derivatives bearing mutations (μ) in two RBPJ consensus motifs (**B**) and a WT *IER5* enhancer D luciferase reporter gene and derivatives bearing mutations in two RBPJ consensus motifs or in flanking AP1 consensus motifs (**C**). Reporter gene assays were performed in SC2 cells maintained in GSI

*Figure 4 continued on next page*

Figure 4 continued

or 24 hr after GSI washout (WO). Luciferase reporter gene activity was determined in biological replicates prepared in triplicate and normalized to the activity of a *Renilla* luciferase internal control gene. Error bars represent standard deviations. (D) Cartoon showing the CRISPR/Cas9 targeting strategy for *IER5* enhancers D and E. (E) Relative *IER5* transcript levels in SC2 cells targeted with control AAVS1 CRISPR/Cas9 plasmids (SC2/con) or with CRISPR/Cas9 plasmids that remove the RPBJ sites in enhancers D and E (SC2/ΔD+E). Cells were either maintained in GSI or were harvested 2 hr following GSI washout (WO). Transcript abundance was measured in experimental triplicates by RT-PCR and normalized against GAPDH. Error bars represent standard errors of the mean. (F) Western blots showing IER5 protein levels in SC2/con cells and SC2/ΔD+E cells that were either maintained in GSI or harvested 1, 2, or 4 hr following GSI washout (WO). In B, C, and E, *, p<0.05; **, p<0.005; ***, p<0.0005 (all two-tailed student t test); NS, not significant.

The online version of this article includes the following figure supplement(s) for figure 4:

**Figure supplement 1.** *IER5* is regulated by Notch in non-transformed keratinocytes.

parasite immune responses in mice (*Kobia et al., 2020*). The Notch target genes and elements identified here in malignant squamous cells are likely to be relevant for understanding Notch function in non-transformed squamous cells, as many of the NTC-binding enhancers found near Notch target genes in SCC cells are also active in normal human keratinocytes (based on review of ENCODE data for non-transformed human keratinocytes). These observations have a number of implications for understanding how Notch regulates the growth and differentiation of squamous cells and highlight the potential for Notch to influence the activity of diverse signaling pathways in keratinocytes through modulation of PP2A.

Prior work has suggested that p53-mediated upregulation of Notch expression and activity is a component of the DNA damage response in keratinocytes (*Mandinova et al., 2008*). Conversely, our work shows that Notch activation, even in a *TP53* mutant background, induces the expression of genes that are components of the keratinocyte DNA damage/cell stress response, suggesting that these two pathways converge on a set of genes that induce the differentiation (and thus, the elimination) of damaged cells. Consistent with this idea, recent work has shown that low-dose radiation induces the differentiation of esophageal squamous cells at the expense of self-renewal in vivo (*Fernandez-Antoran et al., 2019*). The frequent co-mutation of *TP53* and Notch genes in squamous carcinoma may also reflect, at least in part, the convergence of these pathways on a core set of genes with anti-oncogenic activities. Further work delineating the crosstalk between p53 and Notch signaling in well controlled model systems will be needed to test this idea.

Among the genes linked to Notch and p53 is *IER5,* an immediate early response gene that is a component of the DNA-damage response in a number of cell types (*Ding et al., 2009*; *Kis et al., 2006*; *Kumar et al., 1998*; *Yang et al., 2016*). Several lines of investigation suggest that IER5 modulates the function of PP2A complexes containing B55 (*Asano et al., 2016*; *Ishikawa et al., 2015*; *Kawabata et al., 2015*) regulatory subunits, and here we demonstrate that IER5 directly binds B55α protein in a purified system. However, the exact effect of IER5 on PP2A function is uncertain. One model suggests that IER5 augments the ability of B55/PP2A complexes to recognize and dephosphorylate specific substrates such as S6 kinase and HSF1 (*Ishikawa et al., 2015*; *Kawabata et al., 2015*), the latter leading to HSF1 activation as part of the heat shock response. However, our work with double knockout cells suggests IER5 inhibits at least some activities that are attributable to B55α-containing PP2A complexes. Work in purified systems, which is now feasible, may help to clarify how IER5 influences B55/PP2A function.

We also note that *IER3*, a putative regulator of B56/PP2A complexes, behaves as a direct Notch target in our SCC model system. Given the pleotropic role of PP2A isoforms and their myriad substrates, it appears likely that Notch-dependent modulation of PP2A regulators such as IER5 and IER3 will alter the activity of many factors that are regulated by serine/threonine phosphorylation. Because IER5 is required for expression of a large number of Notch-sensitive genes with delayed response kinetics, B55α/PP2A complexes are likely to regulate one or more transcription factors that coordinately induce squamous differentiation with Notch, another idea that is readily testable in our model system.

Finally, we note that our small screen of SCC cell lines suggests that squamous cell carcinomas retain the capacity to respond to Notch signals by undergoing growth arrest and differentiation. Although originally identified as an oncogene, sequencing of cancer genomes has revealed that Notch most commonly acts as a tumor suppressor, particularly in squamous cell carcinoma, which is

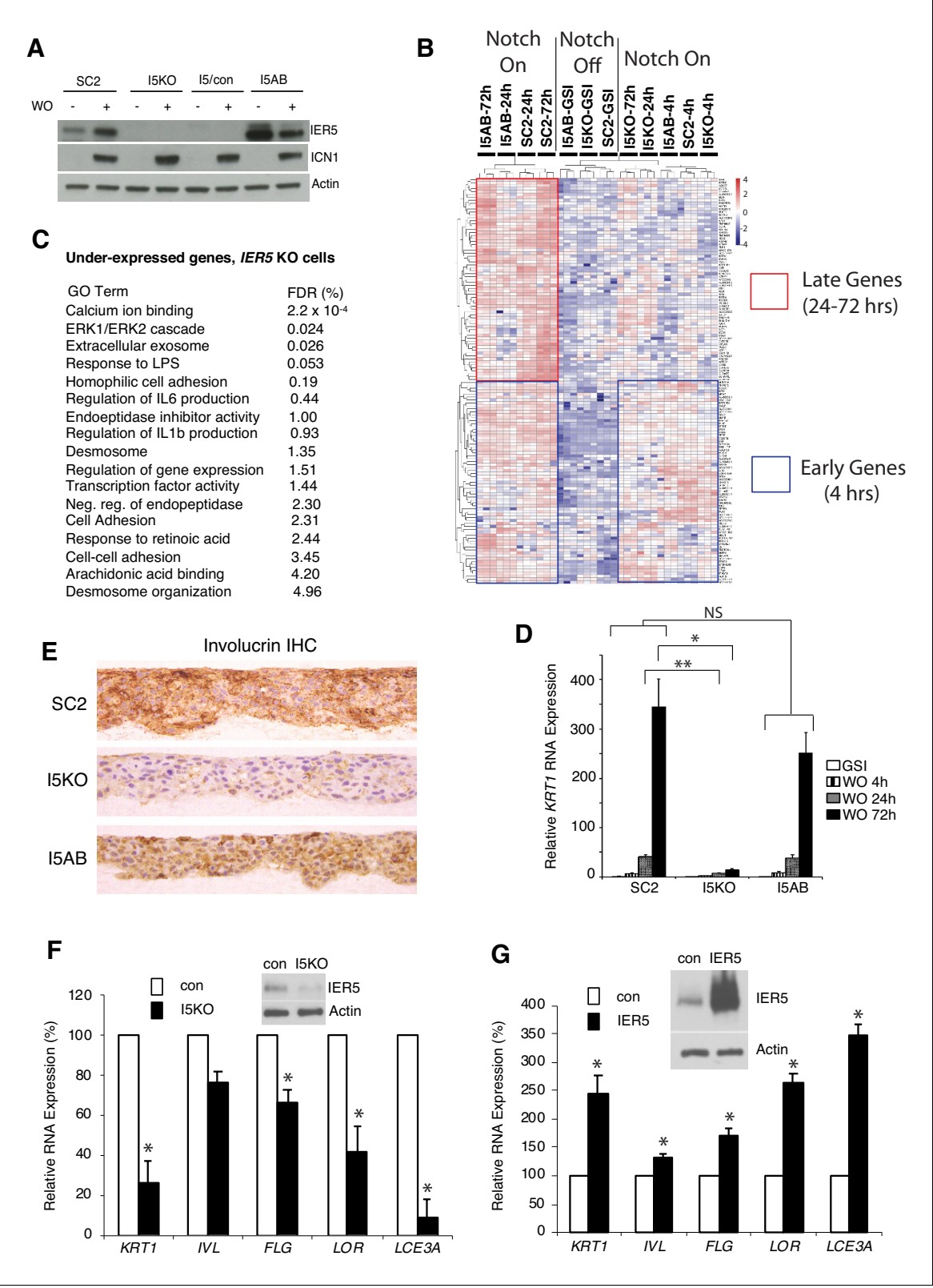

**Figure 5.** Effect of *IER5* on Notch-dependent changes in gene expression in SC2 cells and NOK1 cells. (**A**) Western blots showing IER5 and ICN1 protein levels in SC2 cells, a single-cell clone derived from SC2-*IER5* knockout cells (I5KO), I5KO cells transduced with empty virus (I5/con), and pooled I5KO cells transduced with *IER5* cDNA (I5AB) that were maintained in GSI (-) or harvested 48 hr post-GSI washout (+). (**B**) Heat map showing Notch-induced changes in gene expression in SC2 cells, I5KO cells, and I5AB cells. RNA-seq was performed in biological replicates in triplicate at time 0, 4 hr,

*Figure 5 continued on next page*

*Figure 5 continued*

24 hr, and 72 hr after GSI washout. Samples were subjected to unsupervised clustering using a gene set containing all genes that were significantly upregulated at any time point after Notch activation in SC2 cells. The blue boxes highlight genes that are upregulated at 4, 24, and 72 hr after Notch activation, whereas the red box highlights genes that are under-expressed in *IER5* knockout cells (I5KO) and rescued by re-expression of *IER5* (I5AB) at later timepoints (24 and 72 hr). (C) Gene ontogeny (GO) terms associated with the set of under-expressed genes in I5KO cells following Notch activation. FDR = false discovery rate. (D) Diminished induction of *KRT1* expression at 24 and 72 hr after GSI WO in I5KO cells is prevented by *IER5* addback (I5AB cells). Transcript abundance in biological replicates prepared in triplicate was measured by RT-PCR and normalized against GAPDH. Error bars represent standard deviations of the mean. *, p<0.05; **, p<0.005; NS, not significant (two-tailed student t test). (E) Immunohistochemical staining for involucrin in SC2, I5KO, and I5AB cells in raft cultures grown in the absence of GSI. (F, G) Effect of CRISPR/Cas9 targeting of *IER5* and enforced *IER5* expression on differentiation-associated transcripts in NOK1 cells. In F, NOK1 cells were transduced with CRISPR/Cas9, GFP, and *IER5* (I5KO) gRNA or AAVS1 control (con) gRNA and sorted for GFP positivity. In G, NOK1 cells were transduced with empty GFP-expressing retrovirus (con) or *IER5* and GFP and sorted. In F and G, analyses were done on pooled GFP-positive transductants, which were moved to high Ca2+ medium for 3 days (F) or 5 days (G) prior to harvest. Inset western blots show the extent of *IER5* loss (F) and *IER5* overexpression (G) relative to control cells. In F and G, transcript abundance was measured in biological replicates prepared in triplicate by RT-PCR and normalized against GAPDH. Error bars represent standard deviations of the mean. *, p<0.05, student two-sided t test.

difficult to treat when advanced in stage. While restoring the expression of defective Notch receptors is problematic, detailed analysis of crosstalk between Notch and other pathways may reveal druggable targets leading to reactivation of tumor suppressive signaling nodes downstream of Notch, which would constitute a new rational therapeutic approach for squamous cancers.

# Materials and methods

## Key resources table

| Reagent type (species) or resource | Designation | Source or reference | Identifiers | Additional information |
|---|---|---|---|---|
| Antibody | Rabbit monoclonal anti-MAML1 | Cell Signaling Technology | Cat. #: 12166 | ChIP, 1 ml per $1 \times 10^6$ cells |
| Antibody | Rabbit monoclonal anti-RBPJ | Cell Signaling Technology | Cat. #: 5313 | ChIP, 2.5 ml per $1 \times 10^6$ cells |
| Antibody | Rabbit polyclonal anti-histone H3 acetyl K27 | Abcam | Cat. #: ab4729 | ChIP, 9 ml per $1 \times 10^6$ cells |
| Antibody | Mouse monoclonal anti-involucrin | Sigma | Cat. #: I9018 | IF, 1:500; IHC, 1:10,000 |
| Antibody | Rabbit polyclonal anti-plakophilin-1 | Sigma | Cat. #: HPA027221 | IF, 1:300; IHC, 1:500 |
| Antibody | Rabbit monoclonal anti-activated NOTCH1 (ICN1) | Cell Signaling Technology | Cat. #: 4147 | IHC, 1:50; WB, 1:1000 |
| Antibody | Rabbit monoclonal anti-keratin-1 | Abcam | Cat. #: ab185628 | IHC, 1:1000 |
| Antibody | Rabbit monoclonal anti-Ki-67 | Biocare | Cat. #: CRM325 | IHC, 1:100 |
| Antibody | Mouse monoclonal anti-B55a | Cell Signaling Technology | Cat. #: 5689 | WB, 1:1000 |
| Antibody | Rabbit monoclonal anti-NOTCH2 | Cell Signaling Technology | Cat. #: 5732 | WB, 1:1000 |
| Antibody | Rabbit monoclonal anti-NOTCH3 | Cell Signaling Technology | Cat. #: 5276 | WB, 1:1000 |
| Antibody | Rabbit monoclonal anti-GADD45A | Cell Signaling Technology | Cat. #: 4632 | WB, 1:1000 |
| Antibody | Rabbit monoclonal anti-ID3 | Cell Signaling Technology | Cat. #: 9837 | WB, 1:1000 |

*Continued on next page*

*Continued*

| Reagent type (species) or resource | Designation | Source or reference | Identifiers | Additional information |
|---|---|---|---|---|
| Antibody | Horse polyclonal anti-mouse Ig linked to HRP | Cell Signaling Technology | Cat. #: 7076 | WB, 1:1,000 -1:20,000 |
| Antibody | Goat polyclonal anti-rabbit Ig linked to HRP | Cell Signaling Technology | Cat. #: 7074 | WB, 1:1000 |
| Antibody | Mouse monoclonal anti-actin | Sigma | Cat. #: A1978 | WB, 1:10,000 |
| Antibody | Mouse monoclonal anti-FLAG | Sigma | Cat. #: F3165 | WB, 1:1000 |
| Antibody | Rabbit polyclonal anti-IER5 | Sigma | Cat. #: HPA029894 | WB, 1:1000 |
| Antibody | Mouse monoclonal anti-filaggrin | Santa Cruz Biotechnology | Cat. #: sc-66192 | IHC, 1:100 |
| Antibody | Mouse monoclonal anti-p63 | Biocare Medical | Cat. #: CM163A | IHC, 1:250 |
| Antibody | Rabbit polyclonal anti-loricrin | BioLegend | Cat. #: 905103 | IHC, 1:800 |
| Antibody | Mouse monoclonal anti-BCL6 | Cell Marque Tissue Diagnostics | Cat. #: 227 M-95 | IHC, 1:500 |
| Antibody | Rabbit monoclonal anti-keratin5 | Cell Signaling Technology | Cat. #: 71536 | IHC, 1:2000 |
| Antibody | Chicken polyclonal anti-keratin14 | BioLegend | Cat. #: 906004 | IHC, 1:800 |
| Antibody | Chicken polyclonal anti-SUMO | Lifesensors | Cat. #: AB7002 | WB, 1:2000 |
| Antibody | Sheep polyclonal anti-rabbit Ig linked to Dynabeads | ThermoFisher Scientific | Cat. #: 11203D | ChIP, 100 µl beads per $20 \times 10^6$ cells |
| Antibody | Mouse monoclonal anti-FLAG epitope linked to magnetic beads | Sigma | Cat #: M8823 | Tandem purification, 40 µl to 1 ml beads |
| Cell line (*Homo sapiens*) | IC8 | 10.1038/s41467-018-06027-1 | | Dr. Andrew South (Thomas Jefferson University) |
| Cell line (*H. sapiens*) | SCCT2 | 10.1038/s41467-018-06027-1 | | Dr. Andrew South (Thomas Jefferson University) |
| Cell line (*H. sapiens*) | NOK1 | *Piboonniyom et al., 2003*; 63:476–83 | | Dr. Karl Munger (Tufts University) |
| Commercial assay or kit | CellTiter Blue | Promega | Cat. #: G8080 | |
| Commercial assay or kit | ChIP Assay Kit | Millipore | Cat. #: 17–295 | |
| Commercial assay or kit | Next Ultra II DNA Library Prep Kit | New England BioLabs | Cat. #: E7645 | |
| Commercial assay or kit | Next Ultra II RNA Library Prep Kit | New England BioLabs | Cat. #: E7775 | |
| Commercial assay or kit | QuickChange II Kit | Agilent Technologies | Cat. #: 200523 | |
| Commercial assay or kit | Dual Luciferase Kit | Promega | Cat. #: E1910 | |

*Continued on next page*

*Continued*

| Reagent type (species) or resource | Designation | Source or reference | Identifiers | Additional information |
|---|---|---|---|---|
| Chemical compound, drug | Compound E | Tocris | Cat. #: CAS 209986-17-4 | |
| Recombinant DNA reagent | pL-CRISPR. SFFV.GFP | Addgene | Cat. #: #57827 | |
| Recombinant DNA reagent | pL-CRISPR.SFFV.tRFP | Addgene | Cat. #: #57826 | |
| Recombinant DNA reagent | lentiCRISPRv2 neo | Addgene | Cat. #: 98292 | |
| Recombinant DNA reagent | lentiCRISPRv2 hygro | Addgene | Cat. #: 98291 | |
| Recombinant DNA reagent | pVL1392 | Expression Systems | Cat. #: 91–012 | |

## Cell lines and 2D cultures

Cells were grown under 5% $CO_2$ at 37°C in media supplemented with glutamine and streptomycin/penicillin. IC8 cells (*Wang et al., 2011*) were cultured in Keratinocyte medium as described (*Purdie et al., 2011*). ΔEGF-L1596H cDNA cloned into pBABE-puro was packaged into pseudotyped retrovirus and used to transduce IC8 and SCCT2 cells, which were selected with puromycin (1 μg/ml). In some instances, cells were also transduced with pseudotyped MigRI retrovirus encoding dominant negative MAML1 fused to GFP (*Weng et al., 2003*). Single-cell IC8 cell clones were isolated by limiting dilution. NOK1 cells were grown in keratinocyte-SFM medium supplemented with human EGF and bovine pituitary extract (BPE) (Thermo Fisher Scientific) and induced to differentiate by transfer to Dulbcecco modified Eagle medium (DMEM) containing 10% fetal bovine serum. The identity of IC8 cells and SCCT2 cells was confirmed by detection of cell-line-specific 'private' driver mutations (*Inman et al., 2018*) by NextGen sequencing. The identity of NOK1 cells was confirmed by STR testing (Genetica Cell Line Testing, Case # CX4-007937). Culture cells were tested for mycoplasma periodically using the LookOut Mycoplasma PCR Detection Kit (Millipore Sigma, Cat. #MP0035).

To determine the effect of collagen matrix on ICN1 levels, $5 \times 10^4$ cells were plated in the presence of GSI in standard Corning six-well plates or Corning Biocoat Collagen 6-well plates coated with rat tail collagen 1. After 4 days, cells were subjected to GSI washout or sham GSI washout and cultured for an additional 24 hr.

## Cell growth assays

Cell numbers were estimated using CellTiter Blue (Promega) per the manufacturer's recommendations. Fluorescence was measured using a SpectraMax M3 microplate reader (Molecular Devices).

## Organotypic 3D cultures

3D raft cultures were performed on a matrix containing $5 \times 10^5$ J2 3T3 fibroblast cells and rat collagen as described (*Arnette et al., 2016*). Briefly, rafts were allowed to mature for 6–7 days and then were seeded with $5 \times 10^5$ SCC cells in E-medium in the presence of GSI. After 2 days, rafts were raised to the fluid-air interface, and medium was refreshed + / - GSI every 2 days for a total of 12 additional days.

## Targeted exon sequencing

NGS was performed on IC8 and SCCT2 cell genomic DNA using the 'oncopanel' assay (*Abo et al., 2015*; *Wagle et al., 2012*), which covers 447 cancer genes. Briefly, DNA (200 ng) was enriched with the Agilent SureSelect hybrid capture kit and used for library preparation. Following sequencing (Illumina HiSeq 2500), reads were aligned to human genome GRCh37 (hg19) (*Li and Durbin, 2009*), sorted, duplicate marked, and indexed. Base-quality score calibration and alignments around indels was done with Genome Analysis Toolkit (*DePristo et al., 2011*; *McKenna et al., 2010*). Single-nucleotide variant calls were with MuTect (*Cibulskis et al., 2013*). Copy number alterations were determined using RobustCNV. Structural variants were detected using BreaKmer (*Abo et al., 2015*).

## Preparation of ChIP-Seq and RNA-seq libraries

Chromatin was prepared as described (*Wang et al., 2014*) and was immunoprecipitated with antibodies against MAML1 (clone D3K7B) or RBPJ (clone D10A11, both from Cell Signaling Technology) and Dynabeads bearing sheep anti-rabbit Ig (Thermo Fisher Scientific). H3K27ac ChIPs were prepared using the ChIP assay kit (Millipore) and H2K27ac antibody (ab4729, Abcam). ChIP-seq libraries were constructed using the NEBNext Ultra II DNA Library Prep Kit (New England BioLabs). Total RNA was prepared with Trizol (Life Technology) and RNeasy Mini columns (Qiagen). RNA libraries were constructed using the NEBNext Ultra II RNA Library Prep kit (New England BioLab). ChIP-seq and RNA-seq libraries were sequenced on an Illumina NextSeq 500 instrument. ChIP-seq and RNA-seq data sets are deposited in GEO (https://www.ncbi.nlm.nih.gov/geo/query/acc.cgi?acc=GSE156488).

For *Figure 2—figure supplement 1*, IC8 cells were seeded at $6 \times 10^5$ cells per 10 cm dish. The next morning, media was changed to media containing GSI (Compound E, 1 μM) or an equivalent volume of DMSO. One day after the media change, cells were lifted from the dish with trypsin containing 1 μM GSI or an equivalent volume of DMSO and washed with PBS containing 1 μM GSI or vehicle (DMSO), and $10^6$ cells were pelleted and resuspended in TRIzol (Thermo Fisher). Two biological replicates were collected per condition. RNA Spike-in standards (Invitrogen, 1 μL of 1:10 diluted ERCC) were added to each tube of RNA in TRIzol. Total RNA was extracted by phenol/chloroform with MaXtract tubes (Qiagen). RNA quality was assessed by HS RNA ScreenTape on an Agilent Tape Station. Libraries were constructed using the TruSeq Stranded Total RNA Library Prep Gold kit (Illumina) at the HMS Nascent Transcriptomics Core. Samples were sequenced for paired end reads on the Illumina NovaSeq at the Harvard Bauer Center Sequencing Core, using the S1 Flow Cell and the 100 Cycle Kit. RNA-seq data sets for IC8 cells treated with vehicle or GSI are available at https://www.ncbi.nlm.nih.gov/geo/query/acc.cgi?acc=GSE156624.

## ChIP-seq data analysis

Reads were trimmed with Trim Galore (v.0.3.7 using cutadapt v.1.8), assessed for quality with FastQC (v.0.11.3), and aligned to GRCh38/hg38 with bowtie (v.2.0.0; *Langmead and Salzberg, 2012*). Peaks were identified using MACS2 (v.2.1.1; *Zhang et al., 2008*) and annotated using Homer (v3.12, 6-8-2012; *Heinz et al., 2010*). Peaks mapping to repeats (repeatMasker track, from UCSC) or ENCODE blacklisted regions were removed. Overlapping RBPJ and MAML1 peaks were identified with bedtools intersectBed (v2.23.0; *Quinlan and Hall, 2010*). Motif analysis was performed using MEME-ChIP (*Machanick and Bailey, 2011*). Average signal profiles were generated with ngsplot (*Shen et al., 2014*). RBPJ sequence-paired sites (SPSs) were identified as described (*Severson et al., 2017*). Mixed Gaussian curves were generated in R using the mixtools (v.1.1.0) function (*Benaglia et al., 2009*).

## RNA-seq data analysis

Reads were trimmed as described for DNA reads and aligned to human genome GRCh38/hg38 using gencode release 27 annotations and STAR (v.2.5.3a) (*Dobin et al., 2013*). Raw counts from two sequencing runs were loaded into R (*R Development Core Team, 2014*), summed, and filtered to exclude transcripts with <0.5 reads per million mapped before performing differential expression (DE) analysis with edgeR (v.3.16.5) and RUVSeq (v.1.8.0) (*Risso et al., 2014*). After first-pass DE analysis, a control set of 733 genes with FDR > 0.5 in all pairwise comparisons was used with RUVg (k = 1) to identify unwanted variation. Second-pass edgeR analysis included the RUVg weights in the model matrix. Genes with FDR < 5% and absolute logFC >1 were retained for further analysis. DAVID v.6.8 (Huang da, *Huang et al., 2009*) was used for gene ontology (GO) enrichment analysis of the DE gene lists. EdgeR cpm function with library size normalization and log2 conversion was used to generate expression values, which were displayed using pheatmap (R package version 1.0.8). Other plots were made using in-house R scripts (available upon request).

For *Figure 2—figure supplement 1*, sequencing reads were filtered to retain reads with an average quality score $\geq 20$ and were then mapped to hg38 with Ensembl release 99 annotations using Star version 2.7.0 f. (*Dobin et al., 2013*). Duplicated reads were removed. ERCC spike-in reads were mapped using Bowtie version 1.2.2 (*Langmead et al., 2009*). The percentage of reads mapping to the spike-in was not significantly different between samples. To identify differentially expressed

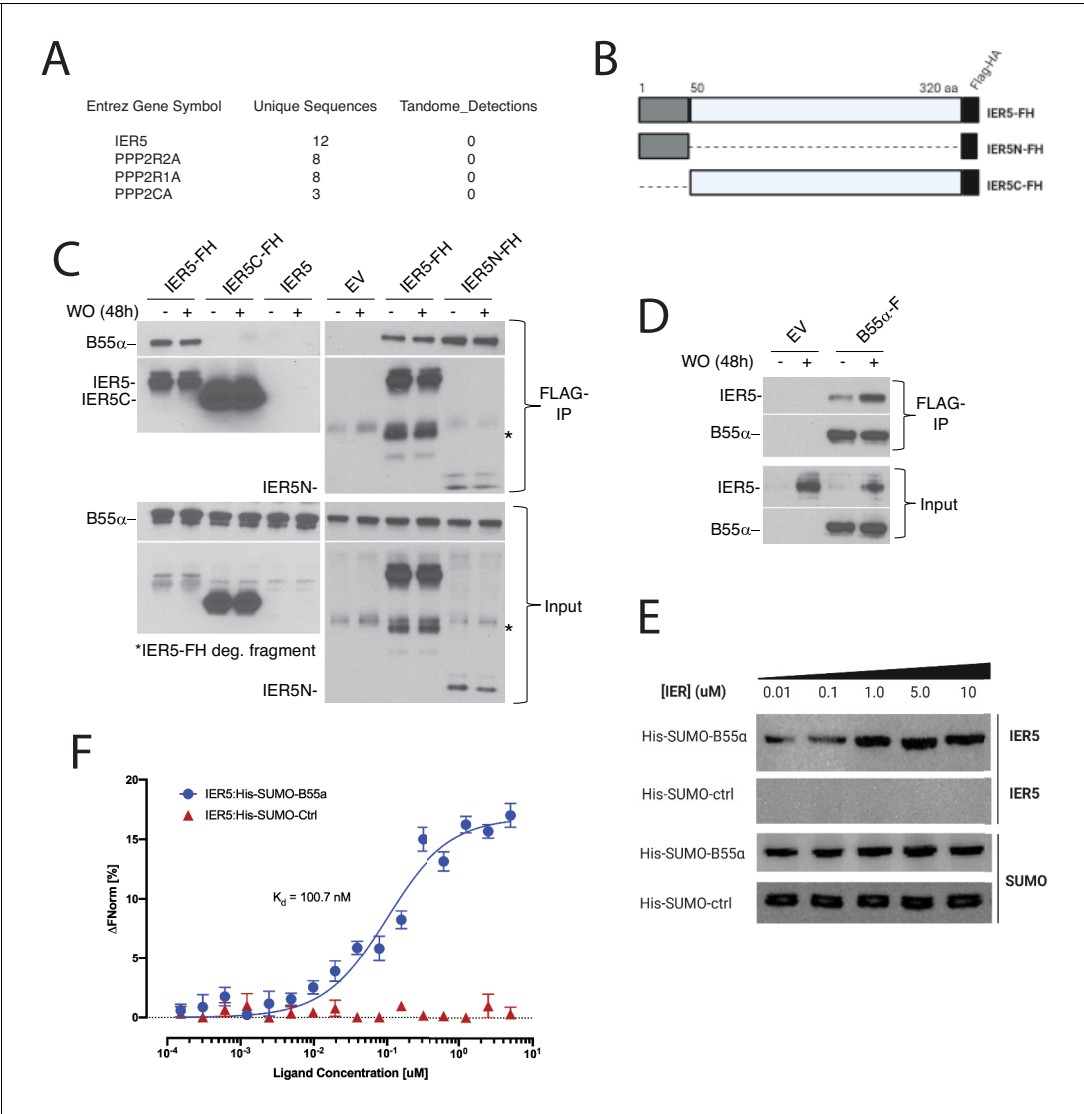

**Figure 6.** IER5 binds to B55α. (**A**) Polypeptides identified by mass spectroscopy in immunoprecipitates prepared from I5 cells expressing tandem-tagged IER5. (**B**) Cartoon showing the structure of tandem-tagged IER5 polypeptides. FH, FLAG-HA tag. (**C**) Western blot analysis of immunoprecipitates prepared from I5 cells expressing the indicated forms of tagged IER5. WO, washout. (**D**) Western blot analysis of immunoprecipitates prepared from SC2 cells expressing FLAG-tagged B55α. (**E**) Western blot showing that IER5 binds His-Sumo-tagged B55α immobilized on beads. The upper two panels were stained for IER5, while the lower two panels were stained for SUMO. (**F**) Microscale thermophoresis showing saturable binding of IER5 to His-Sumo-tagged B55α.

genes, the featureCounts function (*Liao et al., 2014*) in Rsubread version 2.0.1 (*Liao et al., 2019*) was used in R version 3.6.2 to assign reads to Ensembl release 99 gene annotations. Differential gene expression was performed with DESeq2 version 1.26.0 in R (*Love et al., 2014*), using the size factors calculated by DESeq2 to normalize. The cut-offs used to identify differentially expressed genes were an adjusted p-value<0.0001 and |log$_2$ fold change| > 1.

## Quantitative RT-PCR

Total RNA was isolated using RNAeasy Mini Kit (Qiagen) and cDNA was prepared using an iScript cDNA synthesis kit (BioRad). qPCR was carried out using a CFX384 Real-Time PCR Detection System. Gene expression was normalized to GAPDH using the ΔΔ CT method. Primer sets used are available on request.

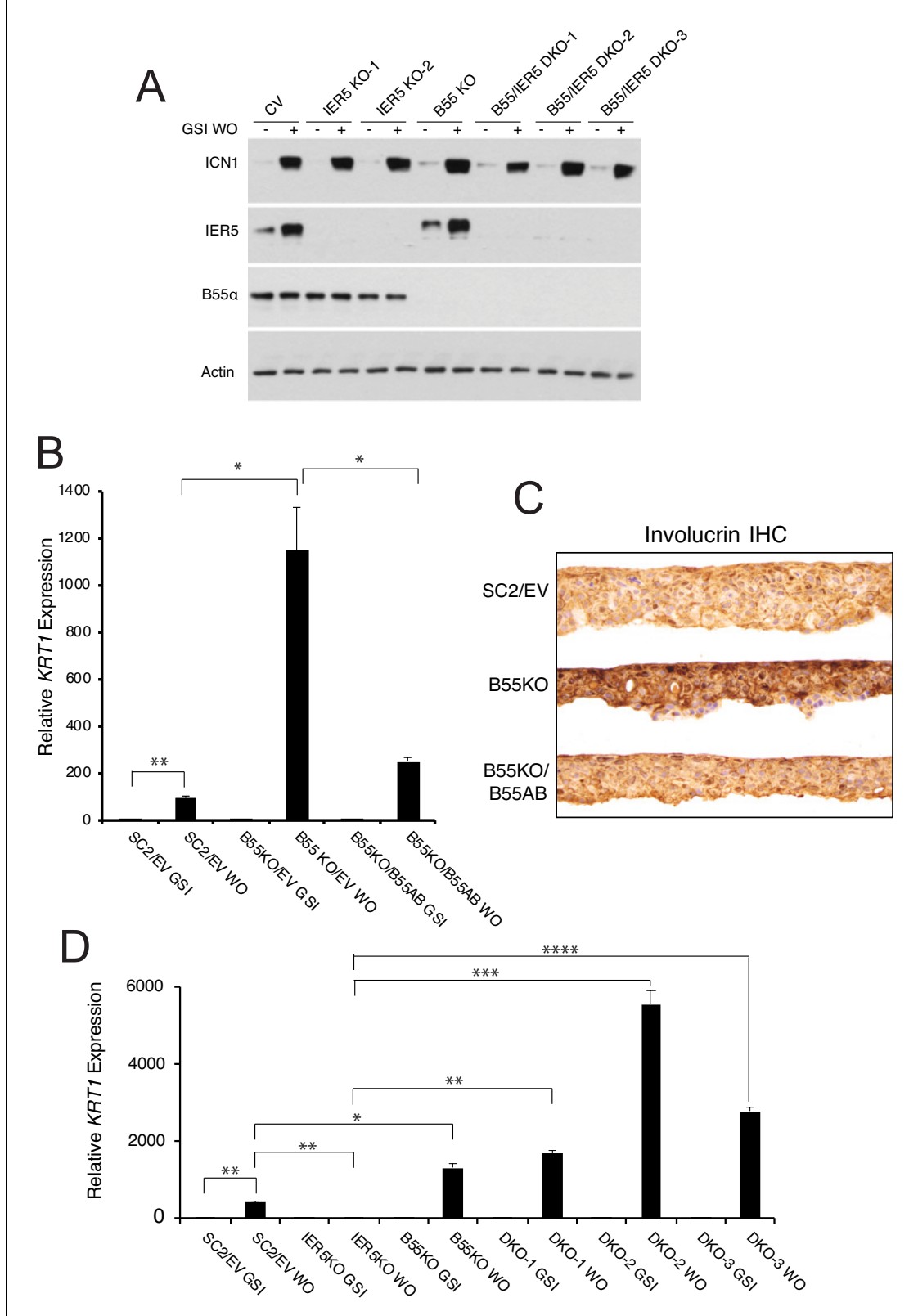

**Figure 7.** *PPP2R2A* is epistatic to *IER5*. (**A**) Western blot showing IER5 and B55α protein levels in single (KO) and double (DKO) *PPP2R2A* and *IER5* knockout clones in the presence of GSI (-) and 4 hr after GSI washout (+). (**B**) *PPP2R2A* knockout enhances Notch-dependent expression of *KRT1*. RT-PCR analysis of *KRT1* expression in SC2 cells transduced with an empty retrovirus (SC2/EV); *PPP2R2A* knockout cells (B55 KO) transduced with empty retrovirus (B55KO/EV); and *PPP2R2A* knockout cells transduced with B55α-expressing retrovirus (B55KO/B55AB). WO = GSI washout. *, p<0.05; **,

*Figure 7 continued on next page*

*Figure 7 continued*

p<0.005 (two-tailed student t test). (**C**) Immunohistochemical (IHC) staining for involucrin of SC2 control, B55KO, and B55KO/B55AB cells in raft cultures in GSI-free medium. (**D**) B55α knockout negates the requirement for *IER5* for Notch-dependent upregulation of *KRT1*. Results are shown for SC2 control cells (SC2/EV); an *IER5* knockout clone; a *PPP2R2A* knockout clone (B55KO); and three *IER5*/*PPP2R2A* double knockout (DKO) clones. Cells were maintained in GSI or harvested 72 hr following GSI washout (WO). *KRT1* transcript abundance was measured in biological replicates prepared in triplicate by RT-PCR and normalized against GAPDH. Error bars represent standard errors of the mean. *, p<0.05; **, p<0.005; ***, p<0.0005; ****, p<0.00005 (all two-tailed student t test).

## In situ hybridization

In situ hybridization (ISH) reagents were from Advanced Cell Diagnostics. Deidentified normal human skin was obtained from the paraffin archives of the Department of Pathology at Brigham and Women's Hospital under institutional review board protocol #2014P001256. Briefly, 4μ sections of skin were deparaffinized, processed using RNAscope 2.5 Universal Pretreatment Reagent, and hybridized to probes specific for human *IER5*, human *PPIB* (peptidylprolyl isomerase B), or bacterial DapB in a HybEZ II oven. ISH signal was developed using the RNAscope 2.5 HD Assay.

## Immunostaining of cells and organotypic rafts

Antibody sources and antibody dilutions are provided in the Key Resource Table. For indirect immunofluorescence microscopy, cells grown on chamber microscope slides were fixed in 4% paraformaldehyde and treated with immunofluorescence blocking buffer (catalog #12411, Cell Signaling Technology). Staining with primary antibodies against involucrin or plakophilin-1 in antibody dilution buffer (catalog #12378, Cell Signaling Technology) was developed by incubation with Alexa Fluor-conjugated secondary antibodies (Cell Signaling Technology, 1:1000). After counterstaining with DAPI (BioLegend, catalog #422801), slides were coverslipped with ProLong Gold Antifade Reagent (Cell Signaling Technology, catalog #9071) and imaged on a Nikon 80i immunofluorescence microscope. Rafts were fixed in 4% buffered formalin for 24 hr, processed, and paraffin-embedded. Sections (4μ) were placed on Superfrost Plus slides and baked at 60°C for 1 hr. Immunohistochemical staining was performed on a Leica Bond III instrument using the following primary antibodies and Leica antigen retrieval conditions: involucrin, retrieval H1 (30 min); plakophilin1, retrieval H1 (30 min); keratin1, H1 retrieval (30 min); keratin5, retrieval H2 (20 min); keratin14, retrieval H2 (20 min); BCL6, retrieval H2 (20 min); p63, retrieval H2 (40 min); filaggrin, retrieval H1 (30 min); loricrin, retrieval H1 (30 min); Ki67, retrieval H2 (20 min); or ICN1, retrieval H2 (40 min). Diaminobenzidine (DAB) staining was developed using the Bond Polymer Refine Detection Kit (Leica). Slides were counterstained with hematoxylin. Digital micrographs were captured with an Olympus BX40 microscope and Olympus cellSens Entry software.

## Reporter gene assays

Luciferase reporter genes containing *IER5*-associated enhancers were assembled in pGL3-TATA (*Wang et al., 2014*). Mutatagenesis was with the QuickChange II kit (Agilent Technologies). Luciferase assays were performed using Dual Luciferase Assay Kit (Promega) as described (*Malecki et al., 2006*) using lysates from cultured cells that were co-transfected with firefly luciferase and internal control *Renilla* luciferase plasmids using Lipofactamine 2000 (Thermo Fisher Scientific).

## Western blotting and immunoprecipitation

Antibody sources and dilutions are provided in the Key Resource Table. Whole cell lysates were prepared as described (*Malecki et al., 2006*). Protein concentration was measured by Bradford assay (Bio-Rad) prior to SDS-PAGE. Western blots staining was developed with Super Signal West Pico Chemiluminescent Substrate (Thermo Scientific). To prepare immunoprecipitates, cells were lysed in 50 mM Tris, pH 7.4, containing 150 mM NaCl, 1 mM EDTA, 1% Triton X-100, and protease inhibitors (Sigma). Lysates were incubated overnight at 4°C with 20 μl anti-FLAG M2 magnetic beads (Sigma). After extensive washing, bound proteins were eluted with FLAG peptide (Sigma) and analyzed on western blots as above.

## Affinity purification of IER5 complexes and mass spectrometry

Mass spectroscopy was performed on tandem affinity purified IER5 complexes prepared from IE5 knockout (I5) cells expressing tandem tagged IER5 48 hr after GSI washout. Lysis of cells and subsequent tandem purification were as described (*Adelmant et al., 2019*). Tryptic peptides were analyzed by electrospray mass spectrometry (QExactive HF mass spectrometer, Thermo Fisher; Digital PicoView electrospray source platform, New Objective). Spectra were recalibrated using the background ion (Si(CH3)2O)six at m/z 445.12 + / - 0.03 and converted to a Mascot generic file format (.mgf) using multiplierz scripts (*Askenazi et al., 2009*; *Parikh et al., 2009*). Spectra were searched using Mascot (v2.6) against three databases: (i) human protein sequences (downloaded from RefSeq); (ii) common lab contaminants; and (iii) a decoy database generated by reversing the sequences from these two databases. Spectra matching peptides from the reverse database were used to calculate a global FDR and were discarded, as were matches to the forward database with FDR > 1.0% and those present in >1% of 108 negative tandem affinity purification controls (*Rozenblatt-Rosen et al., 2012*).

## Recombinant protein expression and purification

A cDNA encoding B55α with 6xHis-SUMO N-terminal tag was cloned into the baculovirus transfer vector pVL1392. High-titer baculovirus supernatants were used to infect insect Sf9 cells grown at a density of $4.0 \times 10^6$ cell/mL. After 72 hr of incubation at 27°C, conditioned media was isolated by centrifugation, supplemented with 20 mM Tris buffer, pH 7.5, containing 150 mM NaCl, 5 mM $CaCl_2$, 1 mM $NiCl_2$, and 0.01 mM $ZnCl_2$, re-centrifuged to remove residual debris, and applied to a Ni-NTA column. After washing with 20 mM Tris buffer, pH 7.5, containing 150 mM NaCl and 5 mM $CaCl_2$, B55α was eluted in the same buffer supplemented with 500 mM imidazole. This eluate was concentrated with a centrifugal filter and then subject to S200 size exclusion chromatography in 20 mM Na cacodylate, pH 6.0, containing 150 mM NaCl, and 5 mM $CaCl_2$. Fractions containing B55α were further purified by ion exchange chromatography on a MonoQ column in 20 mM Na cacodylate, pH 6.0, using a linear NaCl gradient. The purified protein was buffer-exchanged into 20 mM HEPES buffer, pH 7.5, containing 150 mM NaCl, prior to flash freezing and storage at −80°C. A cDNA encoding IER5 with a N-terminal 6xHis-SUMO tag was cloned into pTD6 and used to transform Rosetta *E. coli* cells. Expression of IER5 was induced at 37°C for 4 hr with IPTG induction followed by inclusion body preparation. Lysates were centrifuged at 4°C for 20 min at 30,000 x g and pellets were resuspended in 10 mL wash buffer (50 mM Tris-HCl, pH 7.5, 150 mM NaCl, containing 1% Triton X-100 and 1M urea) per gram cell weight, and incubated at 23°C for 5 min. Following multiple washes, inclusion bodies were resuspended in extraction buffer (50 mM Tris-HCl, pH 7.5, 8M urea, 1 mM β-mercaptoethanol, 1 mM PMSF) and incubated at room temperature for 1 hr. The solubilized proteins were then dialyzed overnight against a 100-fold volume of wash buffer and cleared by centrifugation. 6x-his-IER5 was then concentrated on Ni-NTA beads, eluted with 500 mM imidazole, cleaved using SUMO protease (SUMOpro, Lifesensors), and passed back over Ni-NTA beads. Untagged IER5 was then purified by chromatography on S200 and MonoQ columns as described for B55α.

## IER5/B55α-binding assays

For bead pulldown assays, 5 μM purified His-SUMO-B55α bait was bound to 100 ul Ni-NTA beads, which were mixed with different concentrations of purified IER5 in 20 mM HEPES buffer, pH 7.5, containing 150 mM NaCl for 1 hr. Control binding assays were conducted with purified His-SUMO bound to Ni-NTA beads. Following extensive washing, proteins were eluted by boiling in SDS-PAGE loading buffer and analyzed on SDS-PAGE gels followed by western blotting. Microscale thermophoresis assays were performed on a NanoTemper Monolith NT.115 instrument with blue/red filters (NanoTemper Technologies GmbH, Munich, Germany). Samples were prepared in 20 mM Tris-HCl, pH 7.5, containing 150 mM NaCl, 1 mM β-mercaptoethanol, 1% glycerol, 0.05% Tween-20, and loaded into premium treated capillaries. Measurements were performed at 22° C using 20% MST power with laser off/on times of 5 s and 30 s, respectively. His-SUMO-B55α target labeled with red fluorescent detector as per the NanoTemper His-labeling kit was used at a concentration of 10 nM and mixed with 16 serial dilutions of purified IER5 ligand from 5 μM to 0.000153 μM. All experiments were repeated three times. Data analyses were performed using NanoTemper analysis software.

## CRISPR/Cas9 targeting and site directed mutagenesis

Guide RNAs (gRNAs) were designed using software available at http://crispr.mit.edu. To score the effects of gene editing in bulk cell populations, gRNAs were cloned into pL-CRISPR.SFFV.GFP and transiently transfected using Lipofactamine 2000 (Thermo Fisher Scientific). Cells sorted for GFP expression 48 hr post-transfection were used in downstream analyses. To create double knockout cells, single-cell clones bearing single-gene knockouts were transduced with pL-CRISPR.SFFV.GFP bearing gRNA. Lentivirus was packaged by co-transfection with psPAX2 and pMD2.G. To create deletions, pairs of gRNAs flanking genomic regions of interest were cloned into lentiCRISPRv2 neo and lentiCRISPRv2 hygro, or into pL-CRISPR.SFFV.GFP and pL-CRISPR.SFFV.RFP. Double deletants were isolated sequentially by selection for RFP/GFP double positivity and G418/hygromycin resistance. The sequences of gRNAs used are available on request.

## Acknowledgements

*Funding*: JCA is supported by the Harvard Ludwig Institute and the Michael A Gimbrone Chair in Pathology at Brigham and Women's Hospital and Harvard Medical School. SCB is supported by NIH award R35 CA220340. JMR is a fellow of the Leukemia and Lymphoma Society. *Competing interests*: SCB is on the SAB for Erasca, Inc, receives sponsored research funding from Novartis and Erasca, Inc, and is a consultant for IFM therapeutics and Ayala Pharmaceuticals. JCA is a consultant for Ayala Pharmaceuticals and Cellestia, Inc *Data and materials availability*: All genome-wide data sets are available through GEO. Cell lines, primer sequences, gRNA sequences, and other reagents/methodology will be made available on request.

## Additional information

### Competing interests

Stephen C Blacklow: SCB is on the SAB for Erasca, Inc, receives sponsored research funding from Novartis and Erasca, Inc, and is a consultant for IFM therapeutics and Ayala Pharmaceuticals. Jon C Aster: JCA is a consultant for Ayala Pharmaceuticals and for Cellestia, Inc, There is no conflict of interest with the work described in this manuscript. The other authors declare that no competing interests exist.

### Funding

| Funder | Grant reference number | Author |
| --- | --- | --- |
| Ludwig Institute for Cancer Research | | Jon C Aster |
| National Institutes of Health | R35 CA220340 | Stephen C Blacklow |

The funders had no role in study design, data collection and interpretation, or the decision to submit the work for publication.

### Author contributions

Li Pan, Investigation, Methodology, Writing - original draft, Writing - review and editing, Responsible for conduct of RNA-seq and ChIP-seq experiments, RT-PCR analyses, Western blot analyses, immunofluorescence microscopy, organotypic cultures; Madeleine E Lemieux, Formal analysis, Methodology, Writing - original draft, Writing - review and editing, Responsible for design of RNA-seq experiments, responsible for statistical and bioinformatic analyses of RNA-Seq and ChIP-Seq experiments and generation of graphic displays of data (e.g., heat maps); Tom Thomas, Investigation, Methodology, Writing - original draft, Writing - review and editing, Responsible for purification of recombinant proteins and conduct of tandem affinity purification and immunoprecipitation studies; Julia M Rogers, Investigation, Writing - original draft, Writing - review and editing, Responsible for conducting RNA-seq experiments studying the effects of GSI on gene expression in parental IC8 cells; Colin H Lipper, Investigation, Methodology, Writing - review and editing, Responsible for microscale thermophoresis experiments; Winston Lee, Investigation, Methodology, Writing - original

draft, Writing - review and editing, Responsible for development of organotypic culture system, optimization of the GSI washout method for adherent cell lines; Carl Johnson, Investigation, Writing - original draft, Writing - review and editing, Responsible for western blot analyses, RT-PCR analyses, in situ hybridization studies; Lynette M Sholl, Data curation, Writing - original draft, Writing - review and editing, Responsible for curation of targeted exome sequencing data for the IC8 and SCCT2 cell lines; Andrew P South, Conceptualization, Data curation, Writing - original draft, Writing - review and editing, Assisted with development of organotypic cultures, provided key cell line reagents; Jarrod A Marto, Data curation, Investigation, Methodology, Writing - original draft, Writing - review and editing, Responsible for conduct of tandem mass spectroscopy, curation of data sets to remove non-specific IER5-interacting proteins; Guillaume O Adelmant, Data curation, Formal analysis, Methodology, Writing - original draft, Writing - review and editing, Responsible for conduct of tandem mass spectroscopy, curation of data sets to remove non-specific IER5-interacting proteins; Stephen C Blacklow, Conceptualization, Writing - original draft, Writing - review and editing, Responsible for experimental conceptualization and design; Jon C Aster, Conceptualization, Supervision, Funding acquisition, Methodology, Writing - original draft, Project administration, Writing - review and editing, Responsible for experimental conceptualization and design, documentation of organotypic culture results by light microscopy

### Author ORCIDs
Madeleine E Lemieux (iD) https://orcid.org/0000-0001-6355-8691
Tom Thomas (iD) https://orcid.org/0000-0002-1840-6776
Andrew P South (iD) https://orcid.org/0000-0001-7650-0835
Stephen C Blacklow (iD) https://orcid.org/0000-0002-6904-1981
Jon C Aster (iD) https://orcid.org/0000-0002-1957-9070

### Decision letter and Author response
Decision letter https://doi.org/10.7554/eLife.58081.sa1
Author response https://doi.org/10.7554/eLife.58081.sa2

## Additional files

### Supplementary files
• Supplementary file 1. Differentially expressed genes after 4 hr of Notch activation, SC2 cells.

• Supplementary file 2. Differentially expressed genes after 24 hr of Notch activation, SC2 cells.

• Supplementary file 3. Differentially expressed genes after 72 hr of Notch activation, SC2 cells.

• Supplementary file 4. Unclustered GO annotations of Notch-sensitive genes, SC2 cells, at 72 hr of Notch activation.

• Supplementary file 5. Clustered GO annotations of Notch-sensitive genes, SC2 cells, at 72 hr of Notch activation.

• Supplementary file 6. Comparison of Notch-responsive genes in SC2 cells, MB157 triple-negative breast cancer cells, REC1 mantle cell lymphoma cells, and DND41 T-cell acute lymphoblastic leukemia cells.

• Supplementary file 7. Unclustered GO annotations of *IER5*-dependent Notch-sensitive genes, SC2 cells.

• Supplementary file 8. Clustered GO annotations of *IER5*-dependent Notch-sensitive genes, SC2 cells.

• Supplementary file 9. IER5-interacting proteins identified by tandem affinity purification.

• Transparent reporting form

### Data availability
Sequencing data have been deposited in GEO under accession codes GSE156488 and GSE156624.

The following datasets were generated:

| Author(s) | Year | Dataset title | Dataset URL | Database and Identifier |
|---|---|---|---|---|
| Pan L | 2020 | Notch target genes in keratinocytes | https://www.ncbi.nlm.nih.gov/geo/query/acc.cgi?acc=GSE156488 | NCBI Gene Expression Omnibus, GSE156488 |
| Rogers JM | 2020 | Gamma-secretase inhibition does not affect gene expression in squamous cell carcinoma cells that do not express Notch1 | https://www.ncbi.nlm.nih.gov/geo/query/acc.cgi?acc=GSE156624 | NCBI Gene Expression Omnibus, GSE156624 |

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
