## [Decision Letter]

**Acceptance summary:**

The study combines an elegant approach to control Notch activation in 2D and 3D models of squamous cell differentiation and genome wide analysis of transcriptional programmes, to identify molecules implicated in the DNA damage response and keratinocyte differentiation as downstream targets of Notch signaling. The requirement of a hierarchical signaling cascade, comprising Notch, IER5 (a DNA damage response intermediate) and its association with PP2A, for the regulation of differentiation, is a major insight provided by this work. The study opens new avenues for exploring context dependent signaling of the Notch pathway in other systems.

**Decision letter after peer review:**

Thank you for submitting your article "*IER5*, a DNA-damage response gene, is required for Notch-mediated induction of squamous cell differentiation" for consideration by *eLife*. Your article has been reviewed by two peer reviewers, and the evaluation has been overseen by a Reviewing Editor and Anna Akhmanova as the Senior Editor. The following individuals involved in review of your submission have agreed to reveal their identity: Daniel Schramek (Reviewer #1); Freddy Radtke (Reviewer #2).

The reviewers have discussed the reviews with one another and the Reviewing Editor has drafted this decision to help you prepare a revised submission.

Summary:

In this study, the authors report that Notch signaling in squamous cell carcinomas, induces DNA damage genes, and their implication in keratinocyte differentiation as exemplified by IER5, is novel. The study employs a well-defined model that allows for precise and rapid activation of NOTCH signaling, which is not readily achieved with more conventional methods such as plate-bound ligands. The ChIP seq analysis adds new aspects to how NOTCH regulates differentiation in squamous cells. Overall, the study is interesting, the manuscript is well written and in general the conclusions are justified by the experiments.

Essential revisions:

1) It is unclear why the authors picked IER5 as Notch induced DNA response gene and not any other gene from the family of Notch induced upregulated DNA damage genes. A rationale may be provided.

2) It seems that in the presence of GSI (NOTCH off in all layers), these cells from a stratified epithelium with a basal layer that is negative of differentiation markers (Figure 1F) and suprabasal layers that show pronounced differentiation markers (Figure 1F) but also some proliferating cells (which is a-typical at least in squamous epithelia in vivo). This would indicate that differentiation occurs independent of NOTCH1 (and might be a function of NOTCH2/3). In NOTCH ON conditions (=WO), these cultures still form an epithelial sheet where ICN1 accumulates in the nucleus of suprabasal cells but not in the basal layer. Is this a result of a technical manipulation, or are the cells in contact with the basal lamina down-regulating ICN1 through some post-transcriptional mechanism? From the Materials and methods section, it sounds like that SCC cells where seeded with or without GSI, so 'washout' might be a little misleading. Albeit the highly proliferating nature of the basal layer and absence of ICN1, it seems that these cells at the same time express high levels of differentiation markers such as IVL, PKP1 and Keratin1, which is somehow counterintuitive? So, the major effect of ICN1 in these cultures is the block of proliferation in the suprabasal layers and induction of differentiation markers in the basal (albeit ICN1 negative) layer. An IHC with K5/14, p63, LOR/FLG would explain the observations in light of the known biology of basal versus suprabasal layers and explain/speculate why the basal layer upon WO does not accumulate ICN1, but shows signs of differentiation? K5/K14 staining would further document if the basal layer indeed expresses basal keratinocytes while LOR/FLG will allow assessment of terminal differentiation (which presumably should not overlap with proliferation).

This comment may be addressed including text, clarifying/comparing similarities and differences to epithelial stratification in vivo versus the in vitro system described in this study.

3) The induction of IER5 using an orthogonal approach for NOTCH activation such as EDTA treatment or plate bound ligands and/or short-term NOTCH inhibition in NOTCH1-proficient SCC cells or keratinocytes is critical. This analysis could include RT-PCR for some of newly identified NOTCH targets including IER5 and WB blot for IER5. This will allow the reader to gauge how general the NOTCH targets are over a panel of SCC cells and estimate the level of IER5 transactivation but also expression upon stimulation of endogenous NOTCH1. Figure 5—figure supplement 1A and B strongly suggests that IER5 expression is regulated also by other TF but also potentially post-translationally as the IER5 mRNA levels increase under GSI treatment but the IER5 protein levels do not. This analysis would also give functional insights about the levels of induction of target gene expression.

4) The authors performed ChIP-Seq for RBPJ, MAML1 and H3K27Ac in a system, in which they can express NOTCH1 in a timed control manner. Why have the authors not used antibodies specific for NOTCH1 instead of MAML1? There are several MAML proteins, whereas a ChIP-Seq for NOTCH1 as the authors have done in the past would have been more logical.

5) In Figure 4A ChIP-Seq analysis for RBPJ, MAML and H3K27Ac, within the wider IER5 gene locus is shown. This led to the identification of the putative Notch regulated D and E enhancer. The RBPJ and MAML1 peaks within the D and E region are prominent. Assuming wash out wash out conditions are shown, do these peaks decrease or disappear in the presence of GSI? Are these so called "dynamic" RBPJ sites?

6) The study has successfully identified keratinocyte-specific NOTCH target genes. However, on a global level, it is not entirely clear, which genes are really keratinocyte specifics versus genes that were also found in other systems to be NOTCH targets. As such, a meta-analysis with the few other sets of published NOTCH targets from other systems such as ES cells (Meier-Stiegen et al., 2010 Plos one), myogenic cells (Castel et al., 2013) etc. may be included.

7) To convince the reader that the identified enhancer landscape of the IER5 locus is not a specificity of one particular squamous cell derivative a similar enhancer landscape is shown for non-transformed human keratinocytes (Figure 1—figure supplement 1). The representation is a little odd, the peaks are cut off and could be shown completely.

8) Many graphs lack statistical analysis. For example, CRISPR-mediated deletion of the enhancer region D and E results only in a partial abrogation of NOTCH dependent increase of IER5 transcripts. Is this statistically significant? Similarly, there are no statistics for the luciferase reporter assays (Figure 4B and C) or bar graphs of Figure 5, i.e. in Figure 5—figure supplement 1B, is the Day 1 and Day 5 between DMSO and GSI significantly different?

9) Figure 5F shows apparently a KO of IER5 in NOK1 cells. The band for IER5 is still visible. The KO seems not to be complete. It is at best a knockdown, which is correctly mentioned in the figure legend. Have the authors sequenced the IER locus around the targeting guide RNA? Was only one allele targeted? or is this residual band due to a mixed population of targeted and non-targeted cells.

10) It might be helpful to the inexperienced reader to put *PPP2RA2* and B55a in context and to mention that *PPP2RA2* is the gene coding for the B55a regulatory subunit of PP2A.

11) Although the experiments showing a direct interaction between IER5 and B55a are convincing, the results of the subsequent genetic experiments, do not give any mechanistic insight to how IER5 might regulate squamous cell differentiation through B55a. This part of the story is somewhat confusing and may be better shown as a supplementary figure, concentrating on the identification of IER5 as DNA damage gene involved in regulating squamous cell differentiation.

---

## [Author Response]

Essential revisions:1) It is unclear why the authors picked IER5 as Notch induced DNA response gene and not any other gene from the family of Notch induced upregulated DNA damage genes. A rationale may be provided.

The decision to pursue IER5 as a target was several-fold: (i) it is a novel Notch target; (ii) it is an example of a DNA-damage response gene that is also regulated by Notch; (iii) based on previously published work, it appeared to represent a point of cross-talk between Notch signaling and PP2A, a regulator of many pathways that rely on phosphorylation for signal transmission. We have added these justifications to the subsection “*IER5* is a direct Notch target gene”.

2) It seems that in the presence of GSI (NOTCH off in all layers), these cells from a stratified epithelium with a basal layer that is negative of differentiation markers (Figure 1F) and suprabasal layers that show pronounced differentiation markers (Figure 1F) but also some proliferating cells (which is a-typical at least in squamous epithelia in vivo). This would indicate that differentiation occurs independent of NOTCH1 (and might be a function of NOTCH2/3).

GSI inhibits activation of all four Notch receptors, and as shown later in Figure 5—figure supplement 1, inhibits NOTCH2 and NOTCH3 activation in NOK1 cells; thus, in 3D organotypic cultures when cells are grown at air-fluid interfaces, squamous cells retain some capacity for differentiation that is truly Notch independent (and also p53 independent, since these cells are *TP53* null). We speculate that differentiation may be related to exposure of these cells to high O_2_ tension at the air interface, a condition that is essential to produce epidermis-like growth of normal and transformed keratinocytes in 3D cultures. The mechanism that underlies Notch-independent differentiation of IC8 cells in organotypic cultures is unknown and beyond the scope of the current study. (Parenthetically, our RNA-Seq data show that IC8 cells sharply upregulate NOTCH3 expression in response to GSI washout at 24hr and 72hr of washout, suggesting that as in T-ALL cells, NOTCH3 is a target of NOTCH1 in IC8 cells. By contrast, in IC8 cells NOTCH2 expression did not change significantly at any time point, and NOTCH4 expression was not detectable.)

In NOTCH ON conditions (=WO), these cultures still form an epithelial sheet where ICN1 accumulates in the nucleus of suprabasal cells but not in the basal layer. Is this a result of a technical manipulation, or are the cells in contact with the basal lamina down-regulating ICN1 through some post-transcriptional mechanism?

We also found this observation of great interest, as it suggested that contact of keratinocytes with collagen triggered a signal that destabilized NICD1. Consistent with this idea, in the revised manuscript we show that plating of IC8-DEGF-L1596H cells on collagen (relative to plastic) sharply decreases ICN1 levels without having any effect on total NOTCH1 protein levels. These findings suggest that the organization of IC8-DEGF-L1596H cells into a basal proliferating layer and a suprabasal non-cycling layer stems from a previously unrecognized matrix-dependent effect limits ICN1 accumulation. We show these data in revised Figure 1—figure supplement 1F. Determination of the mechanistic details underlying this effect of matrix clearly merits study but is beyond the scope of the current paper.

From the Materials and methods section, it sounds like that SCC cells where seeded with or without GSI, so 'washout' might be a little misleading.

IC8-DEGF-L1596H cells were indeed plated in the presence of GSI followed by washout or sham washout; the initial description in the Materials and methods was not well worded. We have clarified this point in the Materials and methods section.

Albeit the highly proliferating nature of the basal layer and absence of ICN1, it seems that these cells at the same time express high levels of differentiation markers such as IVL, PKP1 and Keratin1, which is somehow counterintuitive? So, the major effect of ICN1 in these cultures is the block of proliferation in the suprabasal layers and induction of differentiation markers in the basal (albeit ICN1 negative) layer. An IHC with K5/14, p63, LOR/FLG would explain the observations in light of the known biology of basal versus suprabasal layers and explain/speculate why the basal layer upon WO does not accumulate ICN1, but shows signs of differentiation? K5/K14 staining would further document if the basal layer indeed expresses basal keratinocytes while LOR/FLG will allow assessment of terminal differentiation (which presumably should not overlap with proliferation).This comment may be addressed including text, clarifying/comparing similarities and differences to epithelial stratification in vivo versus the in vitro system described in this study.

In our revised paper, we address these comments as follows.

We agree that plakophilin-1 and involucrin staining is not excluded from the basal ICN-low cell layer in the Notch-on state, and as requested by the reviewers now provide a more extensive comparison of the Notch off and on states in IC8 cell rafts with respect to expression of markers that are commonly used to gauge keratinocyte differentiation. Specifically, we have performed staining for p63, K5, K14, BCL6, and FLG and provide representative images of these markers in Figure 1F and Figure 1—figure supplement 3. Notch activation has little/no effect on p63 transcripts (per RNA-seq data), and consistent with these data, in the rafts there is little change in p63 staining following Notch activation. Results with other makers is mixed. On the one hand, K14 staining becomes more sharply restricted to the basal ICN1-low proliferative cells in the Notch-on state (Figure 1F), and Notch activation increases BCL6 levels, particularly in suprabasal cells (Figure 1—figure supplement 3). On the other hand, K5 staining becomes more diffuse through the full thickness of Notch-on rafts, and there is no evidence of induction of markers of terminal differentiation (FLG; Figure 1—figure supplement 3) in the presence or absence of Notch activation. Thus, Notch activation principally alters markers associated with spinous differentiation (as made clear by subsequent RNA-seq analyses) but does not uniformly suppress markers associated with the basal cell state and cannot “push” cells through to terminally differentiate. Since the purpose of our work was to discover direct Notch target genes that may play a role in Notch-dependent progression of basal cells to the spinous cell state, not to determine if Notch activation is sufficient to normalize differentiation of squamous carcinoma cells, we do not think that these are serious discrepancies, although it does highlight the need to evaluate potential target genes and their functional consequences in other models.

3) The induction of IER5 using an orthogonal approach for NOTCH activation such as EDTA treatment or plate bound ligands and/or short-term NOTCH inhibition in NOTCH1-proficient SCC cells or keratinocytes is critical. This analysis could include RT-PCR for some of newly identified NOTCH targets including IER5 and WB blot for IER5. This will allow the reader to gauge how general the NOTCH targets are over a panel of SCC cells and estimate the level of IER5 transactivation but also expression upon stimulation of endogenous NOTCH1.

We have long thought about how best to acutely activate Notch in order to determine its direct effects in cells of various lineages, and for reasons described below, have come to the conclusion that GSI washout has important advantages over other methods. In past work have used GSI washout to identify direct target genes in T-ALL cells (Weng et al., 2006), triple negative breast cancer cells (Cancer Discov 2014; 4:1154-67), and B cell lymphoma cells (Ryan et al., 2017). To the best of our knowledge, because we have rigorously correlated changes in gene expression with the presence of Notch responsive elements that show dynamic changes following Notch activation, observations made in these well controlled systems have stood the test of time. For example, we were one of the first groups to identify MYC as a key Notch target in T-ALL cells using GSI washout (Weng et al., 2006), and subsequently identified the putative pre-T-cell-specific Notch-sensitive *MYC* enhancer ~1.9 MB 3’ of *MYC*, also by doing “dynamic ChIP-Seq” following GSI washout (Wang et al., 2014). In parallel, Adolfo Ferrando’s group did the bold experiment of knocking out this enhancer element and showed that loss of this element produces only two phenotypes in mice: (i) failure of T cell development; and (ii) resistance to Notch-induced T-ALL. Our approach, which identifies both targets and response elements, provides a roadmap for these types of physiologic investigations.

GSI washout coupled with expression of a GSI regulatable form of NOTCH1 to acutely activate Notch is open to two criticisms: (1) the possibility that GSI has off-Notch effects; and (2) that the dose of Notch that is delivered is supraphysiologic and thus may activate target genes that a physiologic dose of Notch would not. With respect to criticism #1, we would like to emphasize the following points. First, although it is certainly true that γ-secretase has many proteolytic targets, we took care to perform RNA-seq on IC8 cells with and without GSI treatment and observed that there were no genes that changed significantly in expression (defined as fold change of 2X and FDR <0.05) following exposure to GSI (Figure 2—figure supplement 1). This fits with the observation that GSI has no effect on the growth or morphology of IC8 cells, even in long term cultures. Second, as an additional new control, we show that the upregulation of *IER5* during the differentiation of NOK1 cells is sensitive to DN-MAML, further implicating endogenous Notch signaling in *IER5* regulation (revised Figure 4—figure supplement 1). Dominant negative MAML1 (DN-MAML) is a highly specific inhibitor of Notch (Weng et al., 2003; Nam et al., 2006) that acts downstream of γ-secretase. Notably, conditional activation of DN-MAML in mice reproduces a host of Notch loss of function phenotypes, including the development of squamous cell carcinoma (Proweller et al., 2006), and has not been observed to produce any off-Notch phenotypes. Taken together, the rapid response of *IER5* to Notch activation (Figure 2E), the identification of Notch-sensitive enhancers flanking *IER5* (Figure 4A), the responsiveness of these elements to Notch in reporter gene assays (Figure 4B, C), the abrogation of Notch-responsiveness following partial CRISPR targeting (Figure 4D-F), and the inhibition of *IER5* upregulation by endogenous Notch signaling in NOK1 cells make a compelling case for *IER5* being a direct Notch target in squamous cells.

By contrast to GSI washout coupled with DN-MAML as an additional control, if one focuses just on the issue of the direct effects of Notch, both EDTA activation of Notch receptors and activation with plated ligand suffer from several limitations. One of the present authors (J.C.A.) was the senior author on the paper that first described EDTA-mediated activation of NOTCH1 and performed many of the experiments in this paper (Rand et al., 2000), and we therefore have a good deal of experience with this approach. In our hands (in unpublished data), all of the Notch activation following EDTA exposure occurs within the first 2-5 minutes of treatment, meaning that this approach generates a relatively small, transient pulse of activated Notch. Shut-off of Notch activation likely stems from chelation of other metals, such as Zn^2+^, that are required for the activity of metalloproteases such as ADAM10 and ADAM17, which in turn act upstream of γ-secretase during Notch activation. That EDTA washout works at all is probably explained by the high affinity of ADAMs for Zn^2+^ relative to the weak binding of Ca^2+^ to the Notch negative regulatory region. Our observations are consistent with those recently published by the Schramek lab, which showed that nuclear NOTCH1 and NOTCH2 appear by 15 minutes following EDTA-treatment of WT murine keratinocytes but then return to near baseline levels by 60 minutes (Supplementary Figures 14 and 15, Loganathan et al., 2020). As a result, only genes that respond rapidly to low levels of activated Notch (and perhaps mainly genes that have low levels of expression prior to EDTA treatment, yielding larger fold changes) will “score” as being responsive in EDTA-treated cells. These limitations may explain why target genes such as *Rhov* (originally identified in the Radtke lab in experiments in normal murine keratinocytes) failed to score in the experiments done by Loganathan et al. Our past observations also highlight a second confounding aspect of EDTA treatment; it has effects on many cellular processes that depend on divalent metal ions, including adhesion to substrate, that undoubtedly affect gene expression independent of effects on Notch. Of the other possible approaches, in principle plated ligand is superior to GSI washout in terms of specificity but plating of adherent cells on ligand-coated substrate is confounded by the need to apply adherent cells that have been suspended following trypsin-EDTA treatment (which activates Notch); moreover, plated ligand lacks the switch-like behavior of GSI washout, which is essential to be able to tightly correlate changes in gene expression with Notch activation. Also, we would like to again highlight the important observation that in the parental cells lacking the Notch transgene, not a single gene shows a significant change in expression in the GSI treated state, using the criteria of FDR<0.05 and log_2_ change of >1 (Figure 2—figure supplement 1). This means that all of the changes that we see in gene expression can be confidently attributed, directly or indirectly, to Notch activation.

The second potential problem with our system is Notch dosage, which could be supraphysiologic and could therefore potentially activate target genes that would not respond to physiologic doses of Notch. Several of the authors (Blacklow and Aster) have in past work systemically compared the activity of various Notch mutations in cultured cells and in vivo. The NRR mutation in the Notch transgene, L1596H, is an example a gain of function mutation of a strength that drives a physiologic response to Notch in murine hematopoietic stem cells in vivo (T cell differentiation from bone marrow progenitors) without causing the development of T cell acute lymphoblastic leukemia (Chiang et al., 2008), a phenotype that does require supraphysiologic doses of Notch. We cannot exclude the possibility that the dose of Notch that we have chosen has non-physiologic effects, but by focusing on genes that respond rapidly we have hopefully kept these to a minimum. In the subsection “Establishment of a Conditional Notch-on SCC Model”, we provide the rationale for the choice of mutant and the use of GSI washout as a method for activating Notch.

We also wish to re-emphasize that we have rigorously shown that endogenous Notch signaling is sufficient to induce *IER5* in immortalized NOK1 keratinocytes. Although *IER5* upregulation in these cells appears to be mediated through NOTCH2 instead of NOTCH1, domain swap experiments in mice have shown that the intracellular domains of these receptors are functionally interchangeable (Liu et al., 2015), and thus all NOTCH1 targets must be regulatable by NOTCH2 as well. In new experiments (Figure 4—figure supplement 1), we show that DN-MAML is a potent inhibitor of IER5 induction via endogenous Notch activation in NOK1 cells, as are other novel targets such as ID3, building additional confidence that the targets that we have focused on are regulated by endogenous, ligand-mediated Notch signaling in keratinocytes.

Figure 5—figure supplement 1A and B strongly suggests that IER5 expression is regulated also by other TF but also potentially post-translationally as the IER5 mRNA levels increase under GSI treatment but the IER5 protein levels do not. This analysis would also give functional insights about the levels of induction of target gene expression.

We agree with the comment that *IER5* regulation is likely to be complex, as Notch inhibition under conditions that induce NOK1 cell differentiation tends to have a greater effect on IER5 protein levels than on *IER5* transcript levels (now shown in revised Figure 4—figure supplement 1). The complex enhancer landscape surrounding *IER5*, the known ability of *TP53* to induce *IER5*, and the existence of multiple phosphorylation sites in IER5 (see https://www.phosphosite.org/proteinAction.action?id=5357431&showAllSites=true) suggest that *IER5* is likely to be regulated by multiple factors at the transcriptional and post-transcriptional level. We have added a description of this complexity to the subsection “*IER5* is a direct Notch target gene”, details of which are beyond the scope of this paper.

4) The authors performed ChIP-Seq for RBPJ, MAML1 and H3K27Ac in a system, in which they can express NOTCH1 in a timed control manner. Why have the authors not used antibodies specific for NOTCH1 instead of MAML1? There are several MAML proteins, whereas a ChIP-Seq for NOTCH1 as the authors have done in the past would have been more logical.

There are several logical reasons why we decided to do ChIP-Seq for MAML1 rather than NOTCH1, as we have done in the past, or other MAMLs. As for the choice of MAML1, RNA-seq data showed that it is the major member of the Mastermind family expressed in IC8 cells (MAML1 log_2_ read counts per million = 6.66; log_2_ MAML2 read counts per million = 3.59; MAML3 log_2_ read counts per million = 0.70). As for not doing ChIP-Seq for NOTCH1, we were driven by the desire to look at complexes containing endogenous Notch transcription complex components (rather than proteins driven from transgenes) and, for the sake of rigor and reproducibility, to perform our analyses with commercial monoclonal antibodies, instead of private polyclonal antibodies [the only NOTCH1 reagent that we have found to produce robust NOTCH1 ChIP-Seq results]. Moreover, given reports of MAML1 participation in transcription complexes other than Notch transcription complexes, we were curious to see if we would detect evidence of such complexes in squamous cells – we did not. We now provide our reasoning in the subsection “Notch target genes are associated with lineage-specific NTC-binding enhancer elements”.

5) In Figure 4A ChIP-Seq analysis for RBPJ, MAML and H3K27Ac, within the wider IER5 gene locus is shown. This led to the identification of the putative Notch regulated D and E enhancer. The RBPJ and MAML1 peaks within the D and E region are prominent. Assuming wash out wash out conditions are shown, do these peaks decrease or disappear in the presence of GSI? Are these so called "dynamic" RBPJ sites?

RBPJ peaks generally do not disappear in the presence of GSI/absence of Notch signals, but as shown in Figure 3C, do on average increase in size in the presence of active Notch. In our experience, the most sensitive measure of “dynamic sites” that are associated with increased transcription of flanking genes is increased H3K27ac, as shown in Figure 4. We now make this point in the subsection “*IER5* is a direct Notch target gene”.

6) The study has successfully identified keratinocyte-specific NOTCH target genes. However, on a global level, it is not entirely clear, which genes are really keratinocyte specifics versus genes that were also found in other systems to be NOTCH targets. As such, a meta-analysis with the few other sets of published NOTCH targets from other systems such as ES cells (Meier-Stiegen et al., 2010 Plos one), myogenic cells (Castel et al., 2013) etc. may be included.

The data in the papers listed above were obtained at different time points under different conditions of Notch activation. A more apt comparison is the list of genes that are rapidly and significantly upregulated by Notch activation (defined as FDR<0.05 and log2 fold change >1) via GSI washout. We have compared this list of genes in squamous carcinoma cells, in which Notch behaves as a tumor suppressor, and in the triple negative breast cancer cell line MB157, the mantle cell lymphoma cell line REC1, and the T-cell acute lymphoblastic cell line DND41, all cell contexts in which Notch has oncogenic activities. In new Supplementary file 6 and Figure 2—figure supplement 2, we show that only around 10% of genes that are acutely regulated by Notch in squamous cells overlap with those that are upregulated in triple negative breast cancer cells, and that less than 2% of Notch responsive genes overlap between squamous cells and blood cancer cells. Moreover, across these four cell types, there are no target genes that are shared. These comparisons, now discussed in the subsection “Identification of a squamous cell-specific Notch-induced program of gene expression”, serve to emphasize the remarkable context specificity of Notch effects, which underlie its ability to produce widely varied cellular outcomes in different cell types.

7) To convince the reader that the identified enhancer landscape of the IER5 locus is not a specificity of one particular squamous cell derivative a similar enhancer landscape is shown for non-transformed human keratinocytes (Figure 1—figure supplement 1). The representation is a little odd, the peaks are cut off and could be shown completely.

We now show the tops of the peaks in revised Figure 4—figure supplement 1.

8) Many graphs lack statistical analysis. For example, CRISPR-mediated deletion of the enhancer region D and E results only in a partial abrogation of NOTCH dependent increase of IER5 transcripts. Is this statistically significant? Similarly, there are no statistics for the luciferase reporter assays (Figure 4B and C) or bar graphs of Figure 5, i.e. in Figure 5—figure supplement 1B, is the Day 1 and Day 5 between DMSO and GSI significantly different?

We now provide p-values in all of these figures.

9) Figure 5F shows apparently a KO of IER5 in NOK1 cells. The band for IER5 is still visible. The KO seems not to be complete. It is at best a knockdown, which is correctly mentioned in the figure legend. Have the authors sequenced the IER locus around the targeting guide RNA? Was only one allele targeted? or is this residual band due to a mixed population of targeted and non-targeted cells.

The latter is the case. This is a bulk population of CRISPR targeted cells and there is a subpopulation of cells that is not successfully targeted; hence, although the effect of IER5 targeting is clear, Figure 5F may underestimate the true size of this effect. We now describe this limitation more clearly in the subsection “*IER5* is required for “late” Notch-dependent differentiation events in squamous cells” and in the figure legend.

10) It might be helpful to the inexperienced reader to put PPP2RA2 and B55a in context and to mention that PPP2RA2 is the gene coding for the B55a regulatory subunit of PP2A.

We have explained that B55a is encoded by the *PPP2RA2* gene in the subsection “*IER5* binds B55α/PP2A complexes”.

11) Although the experiments showing a direct interaction between IER5 and B55a are convincing, the results of the subsequent genetic experiments, do not give any mechanistic insight to how IER5 might regulate squamous cell differentiation through B55a. This part of the story is somewhat confusing and may be better shown as a supplementary figure, concentrating on the identification of IER5 as DNA damage gene involved in regulating squamous cell differentiation.

We appreciate the reviewer’s comments, but respectfully disagree about whether the results are confusing. Our data provide additional evidence that the functional effects of IER5 are linked to B55a, and moreover suggest that these effects are mediated through inhibition of B55a/PP2A complexes. Given uncertainty about the effects of IER5 on B55a/PP2A, we would prefer to retain these results as a figure rather than a supplementary figure.